# Multi-ancestry genome-wide association meta-analysis of Parkinson's disease

Jonggeol Jeffrey Kim [1,2,167] ✉, Dan Vitale[1,3,4,167], Diego Véliz Otani[5,6,167], Michelle Mulan Lian[7,8,167], Karl Heilbron[9], the 23andMe Research Team*, Hirotaka Iwaki[1,3,4], Julie Lake[1], Caroline Warly Solsberg [10,11,12], Hampton Leonard[1,3,4], Mary B. Makarious [1,13,14], Eng-King Tan [15], Andrew B. Singleton[1,4], Sara Bandres-Ciga[1,4], Alastair J. Noyce[2], the Global Parkinson's Genetics Program (GP2)*, Cornelis Blauwendraat [1,4,168] ✉, Mike A. Nalls [1,3,4,168] ✉, Jia Nee Foo[7,8,168] ✉ & Ignacio Mata[16,168] ✉

Although over 90 independent risk variants have been identified for Parkinson's disease using genome-wide association studies, most studies have been performed in just one population at a time. Here we performed a large-scale multi-ancestry meta-analysis of Parkinson's disease with 49,049 cases, 18,785 proxy cases and 2,458,063 controls including individuals of European, East Asian, Latin American and African ancestry. In a meta-analysis, we identified 78 independent genome-wide significant loci, including 12 potentially novel loci (*MTF2*, *PIK3CA*, *ADD1*, *SYBU*, *IRS2*, *USP8*, *PIGL*, *FASN*, *MYLK2*, *USP25*, *EP300* and *PPP6R2*) and fine-mapped 6 putative causal variants at 6 known PD loci. By combining our results with publicly available eQTL data, we identified 25 putative risk genes in these novel loci whose expression is associated with PD risk. This work lays the groundwork for future efforts aimed at identifying PD loci in non-European populations.

Parkinson's disease (PD) is a neurodegenerative disease pathologically defined by Lewy body inclusions in the brain and the death of dopaminergic neurons in the midbrain. The identification of genetic risk factors is imperative for mitigating the global burden of PD, one of the fastest growing age-related neurodegenerative diseases. A large

PD genome-wide association study (GWAS) meta-analysis uncovered 90 independent genetic risk variants in individuals of European ancestry[1]. Similarly, large-scale PD GWAS meta-analyses of East Asian[2] and a single GWAS of Latin American[3] individuals have each identified two risk loci that were not previously identified in Europeans. For PD, there

[1]Laboratory of Neurogenetics, National Institute on Aging, National Institutes of Health, Bethesda, MD, USA. [2]Preventive Neurology Unit, Centre for Prevention Diagnosis and Detection, Wolfson Institute of Population Health, Queen Mary University of London, London, UK. [3]Data Tecnica International, Washington, DC, USA. [4]Center for Alzheimer's and Related Dementias (CARD), National Institute on Aging and National Institute of Neurological Disorders and Stroke, National Institutes of Health, Bethesda, MD, USA. [5]Neurogenetics Research Center, Instituto Nacional de Ciencias Neurológicas, Lima, Peru. [6]Institute for Genome Sciences, University of Maryland, Baltimore, MD, USA. [7]Lee Kong Chian School of Medicine, Nanyang Technological University Singapore, Singapore, Singapore. [8]Genome Institute of Singapore, Agency for Science, Technology and Research, A*STAR, Singapore, Singapore. [9]23andMe, Inc., Sunnyvale, CA, USA. [10]Pharmaceutical Sciences and Pharmacogenomics, UCSF, San Francisco, CA, USA. [11]Department of Neurology and Weill Institute for Neurosciences, University of California, San Francisco, San Francisco, CA, USA. [12]Memory and Aging Center, UCSF, San Francisco, CA, USA. [13]Department of Clinical and Movement Neurosciences, UCL Queen Square Institute of Neurology, London, UK. [14]UCL Movement Disorders Centre, University College London, London, UK. [15]Department of Neurology, National Neuroscience Institute, Duke NUS Medical School, Singapore, Singapore. [16]Genomic Medicine, Lerner Research Institute, Cleveland Clinic Foundation, Cleveland, OH, USA. [167]These authors contributed equally: Jonggeol Jeffrey Kim, Dan Vitale, Diego Veliz-Otani, Michelle Mulan Lian. [168]These authors jointly supervised this work: Cornelis Blauwendraat, Mike A. Nalls, Jia Nee Foo, Ignacio Mata. *A list of authors and their affiliations appears at the end of the paper. ✉e-mail: kimjoj@nih.gov; cornelis.blauwendraat@nih.gov; mike@datatecnica.com; jianee.foo@ntu.edu.sg; matai@ccf.org

are now large-scale efforts to sequence and analyze genomic data in underrepresented populations with the goal of both identifying novel associated loci, fine-mapping known loci and addressing the inequality that exists in current precision medicine efforts[4,5]. Here we performed a large-scale multi-ancestry meta-analysis (MAMA) of PD GWASs by including individuals from four ancestral populations: European, East Asian, Latin American and African. This effort can serve as a guide for future genetic analyses to increase ancestral representation.

## Meta-analyses identify 66 known and 12 novel loci

In addition to results from previously described European[1], East Asian[2] and Latin American[3] studies, we also used FinnGen and additional datasets for East Asian, Latin American and African cohorts from 23andMe, Inc (Table 1, Fig. 1 and Supplementary Table 1). In total, we included 49,049 PD cases, 18,618 proxy cases (first-degree relative with PD) and 2,458,063 neurologically-healthy controls. Genetic covariance intercepts from linkage disequilibrium (LD) score regression[6] within ancestries were close to zero or near the 95% confidence interval, implying that there is no sample overlap between the cohorts (Supplementary Table 1). After the data were harmonized and mapped to genome build hg19, MAMAs were conducted using a random-effects model and meta-regression of multi-ethnic genetic association (MR-MEGA)[7]. The random-effects model had greater power to detect homogenous allelic effects[7]. MR-MEGA uses axes of genetic variation as covariates in its meta-regression analysis and had greater power to detect heterogeneous effects across the different cohorts. MR-MEGA also distinguishes ancestral heterogeneity (differences in effect estimates due to ancestry-level genetic variation) from residual heterogeneity using axes of genetic variation generated from the allele frequencies across the different cohorts.

Combining results from the random-effects model and MR-MEGA, we found 12 novel PD risk loci and 66 hits in known risk loci from single-ancestry GWAS (Table 2, Fig. 2 and Supplementary Tables 2–5) that met the Bonferroni-corrected alpha of $5 \times 10^{-9}$, a more stringent threshold chosen to account for the larger number of haplotypes resulting from the ancestrally diverse datasets[8]. Of the 78 risk loci identified, 69 were significant in the random-effects model, whereas 3 were only significant in MR-MEGA. Eight of the novel loci found by the random-effect method showed homogeneous effects across the four different ancestries. An additional novel locus (FASN) identified by the random-effect method showed homogeneous effects in all available populations, but note that this variant failed quality control in both East Asian datasets. The other three loci, identified exclusively in MR-MEGA, showed ancestrally heterogeneous effects. All three loci (IRS2, MYLK2 and USP25) showed evidence of significant ancestral heterogeneity ($P_{\text{ANC-HET}} < 0.05$) but no significant residual heterogeneity ($P_{\text{RES-HET}} > 0.148$), supporting the idea that the signals are due to population structural differences rather than other confounding factors (Fig. 3). For the IRS2 locus (lead SNP rs1078514, $P_{\text{ANC-HET}} = 5.3 \times 10^{-3}$) the Finnish cohort has an opposite effect direction compared to the meta-analysis effect estimate (Supplementary Fig. 4). Similarly, the MYLK2 locus has the African effect estimate most different from the meta-analysis effect estimate (lead SNP rs6060983, $P_{\text{ANC-HET}} = 0.035$), suggesting different effects between populations. Although this is a novel single-trait GWAS locus, its lead SNP was previously discovered as a potential pleiotropic locus in a multi-trait conditional/conjunctional false discovery rate (FDR) study between schizophrenia and PD[9]. Lastly, the USP25 locus had the most significant ancestral heterogeneity (lead SNP rs1736020, $P_{\text{ANC-HET}} = 4.74 \times 10^{-5}$) and its effects were specific to European and African cohorts, albeit in different directions. When looking at the nearest protein coding gene to each novel lead SNP and their probability of being loss-of-function intolerant (pLI) score, we found that 7 out of 12 genes had a pLI score of 0.99 or 1. Genes with low pLI scores were found both in loci with (MYLK2) and without (SYBU, PIGL and PPP6R2) significant ancestry heterogeneity.

**Table 1 | Cohort descriptions**

| Study | Ancestral population | Cases/proxy/controls |
|---|---|---|
| Nalls et al.[1] | European (EUR) | 37,688/18,618/1,411,006 |
| Foo et al.[2] | East Asian (EAS) | 6,724/0/24,851 |
| LARGE-PD 3 | Latin American (AMR) | 807/0/690 |
| FinnGen Release 4 | European-Finnish (EUR) | 1,587/0/94,096 |
| 23andMe—African | African (AFR) | 288/0/193,985 |
| 23andMe—East Asian | East Asian (EAS) | 322/0/151,905 |
| 23andMe—Latino | Latin American (AMR) | 1,633/0/581,530 |
| MAMA | | 49,049/18,618/2,458,063 |

PESCA v0.3 (ref. 10) was run for the main European and East Asian meta-analyses and all loci identified in the main analysis were explored (Supplementary Table 6). PESCA uses ancestry-matched LD estimates to infer whether the causal variants are population-specific or shared between two populations. Variants identified as shared between the populations may be more likely to be causal. In addition, we expect higher posterior probability (PP) for shared causal variants in the loci identified by MAMA, even if they have not previously been identified in the single-ancestry study. The lead SNP in the RIMS1 locus (rs12528068) had a high PP for being a shared causal variant (PP = 0.972) despite being significant in the European study[1] but not in the East Asian study[2]. We also observed that the novel lead variants for MTF2 (rs35940311), PIK3CA (rs11918587), EP300 (rs4820434) and PPP6R2 (rs60708277) had higher PP estimates for being shared causal variants across both populations ($PP_{\text{shared}} = 0.757, 0.214, 0.769, 0.946$) than for being causal variants in a single population ($PP_{\text{EUR}} < 0.080$, $PP_{\text{EAS}} < 0.001$). However, it is important to note that the sample size discrepancy between the European and East Asian data impacts our power to detect population-specific causal variants at any of these loci.

We found 17 suggestive loci that failed to meet our stringent significance threshold but had $P < 5 \times 10^{-8}$ in a fixed-effects meta-analysis and $P < 1 \times 10^{-6}$ in the random-effects meta-analysis (Supplementary Table 4). Fourteen of these regions were novel loci. Two loci near JAK1 and HS1BP3 were exclusively found in the 23andMe Latin American and African cohorts. The lead SNPs (rs578139575 and rs73919910) for these loci are non-coding and very rare in European populations but are more common in Africans and Latin Americans (gnomAD v3.1.2 minor allele frequencies in EUR: 0.02%, 0.23%; AFR: 1.64%, 8.84%; AMR: 0.41%, 1.91%). If confirmed, these loci would confer a strong effect on PD risk (beta: −1.3, −0.54). These loci merit further studies in the African and Latin American populations.

## Fine-mapping identifies six credible sets with single variants

Fine-mapping was also performed using MR-MEGA, which uses ancestry heterogeneity to increase fine-mapping resolution. We identified 23 loci that had fewer than 5 variants within the 95% credible set. Of these, MR-MEGA nominated a single putative causal variant with >95% PP in 6 loci: TMEM163, TMEM175, SNCA, CAMK2D, HIP1R and LSM7 (Table 3 and Supplementary Tables 7 and 8). Our results affirmed previous results showing the TMEM175 p.M393T coding variant as the likely causal variant[11]. The putative variants HIP1R have strong evidence for regulome binding (RegulomeDB rank ≤ 2). In particular the HIP1R variant rs10847864 is located in a transcription start site that is active in substantia nigra tissue (chromatin state windows: chr12:123326200.123327200) and astrocytes in the spinal cord and the brain (chromatin state windows: chr12:123326400.123326600). Outside of the credible sets containing a single variant, we identified missense variants in two genes: FCGR2A (p.H167R, PP = 0.145) and SLC18B1 (p.S30P, PP = 0.780).

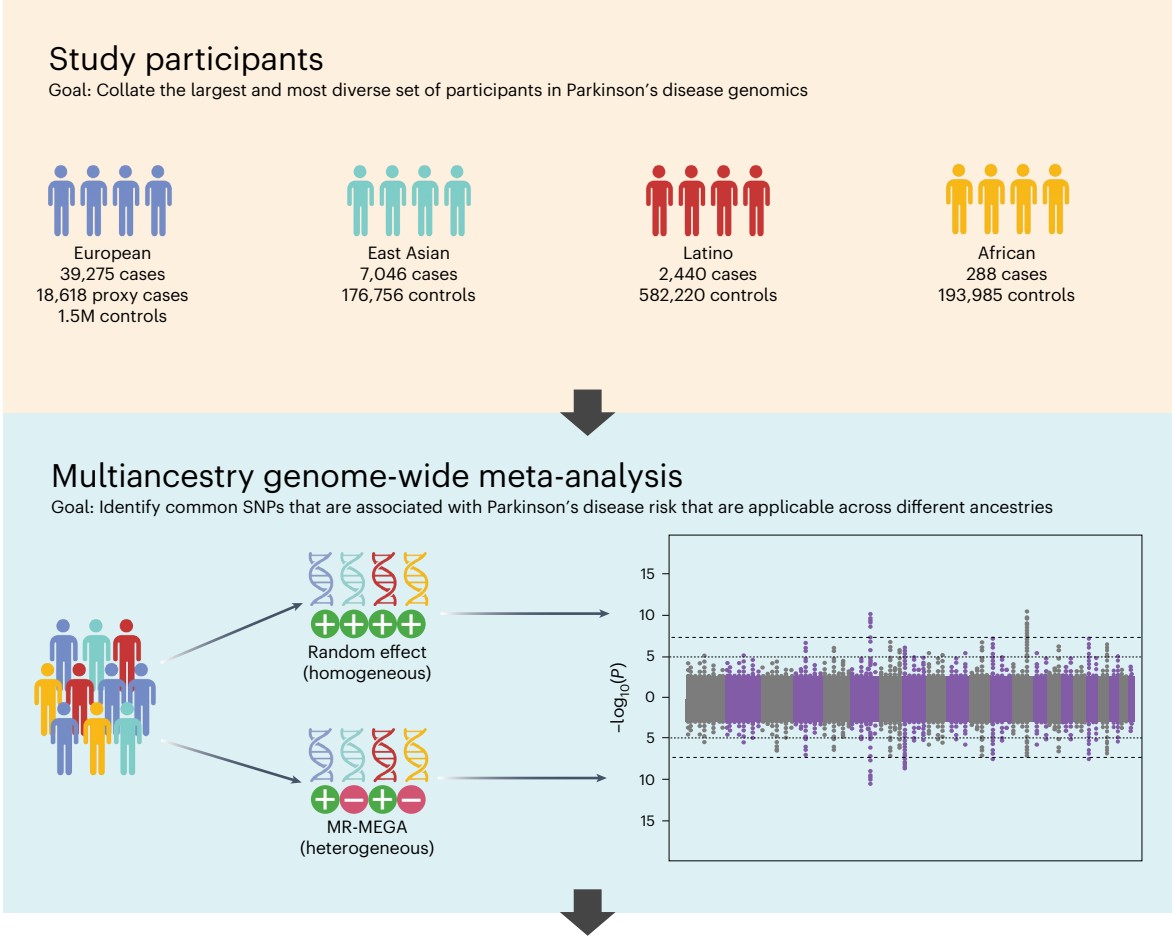

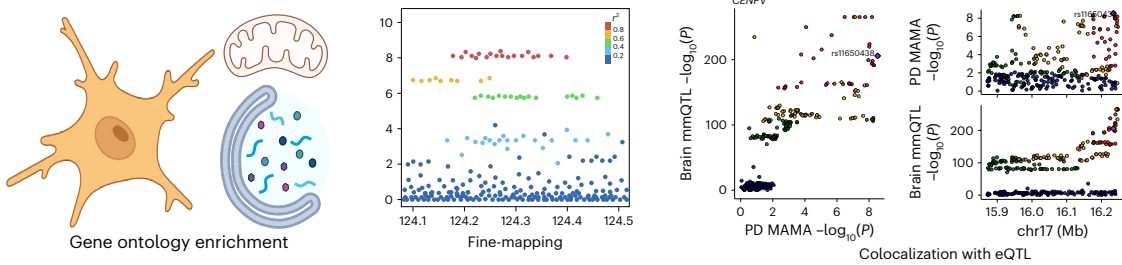

**Fig. 1 | MAMA study design.** Top panel: four ancestry groups used in the meta-analysis. Middle panel: MAMA and the two methods used. Random-effect (top) is better suited for risk variants with homogeneous effect direction across different ancestries, whereas MR-MEGA (bottom) can identify risk variants with heterogeneous effects due to population stratification introduced by ancestry differences. The densely dashed lines indicate Bonferroni adjusted suggestive threshold of two-sided $P < 1 \times 10^{-6}$, and the loosely dashed lines indicate Bonferroni adjusted significant threshold of two-sided $P < 5 \times 10^{-9}$. Bottom panel: downstream analyses and their examples. Created with Biorender.com.

## Gene set analysis finds enrichment in brain tissues

We used the Functional Mapping and Annotation (FUMA) software[12,13] to functionally annotate the random-effect results. We generated a custom 1000 Genome reference panel that reflected the ancestry proportions of our dataset and ran multi-marker analysis of genomic annotation (MAGMA)[14] for gene ontology, tissue level and single-cell expression data. We tested 16,992 gene ontology sets in MSigDB v7.0 (ref. 15) and used conditional analysis to discard redundant terms or identify gene sets that must be interpreted together. We found that 40 gene sets were significantly enriched with conditional analysis

identifying 13 gene sets that share their signals with at least one other gene set (Supplementary Table 9). This is a substantial increase from previous 10 gene sets in the European meta-analysis performed by Nalls and colleagues[1]. Only two gene ontology terms that were significant in the Nalls et al. meta-analysis were also significant in the multi-ancestry results after multiple test correction: 'curated geneset: Ikeda MIR30 Targets Up' ($P_{FDR} = 0.018$) and 'cellular component: vacuolar membrane' ($P_{FDR} = 0.047$). In addition, ontology terms in immune system pathways (microglial cell proliferation, macrophage proliferation, natural killer T cell differentiation: $P_{FDR} < 0.04$), mitochondria (response to mitochondrial depolarization: $P_{FDR} = 0.028$), vesicles (vesicle uncoating,

**Table 2 | Meta-analysis results of lead SNPs in the novel loci**

| rsID | Nearest coding gene | SMR nominated putative genes | CHR:BP:A1:A2 | BETA(RE) | SE | P(RE) | P(MR-MEGA) | P(ANC-HET) | P(RES-HET) | gnomAD EUR AF | gnomAD EAS AF | gnomAD AMR AF | gnomAD AFR AF | pLI |
|---|---|---|---|---|---|---|---|---|---|---|---|---|---|---|
| rs11164870 | MTF2 | CCDC18 | 1:93552187:C:G | 0.054 | 0.009 | $1.15\times10^{-10}$ | $2.64\times10^{-9}$ | 0.229 | 0.928 | 39.0% | 35.1% | 45.2% | 85.0% | 1 |
| rs6806917 | PIK3CA | KCNMB3 | 3:178861417:T:C | −0.070 | 0.011 | $1.65\times10^{-10}$ | $3.43\times10^{-9}$ | 0.215 | 0.762 | 82.0% | 89.9% | 77.5% | 57.8% | 1 |
| rs16843452 | ADD1 | ADD1, NOP14-AS1, NOP14 | 4:2849168:T:C | −0.068 | 0.012 | $4.11\times10^{-9}$ | $3.19\times10^{-7}$ | 0.747 | 0.687 | 18.5% | 47.4% | 18.2% | 8.9% | 0.99 |
| rs6469271 | SYBU | SYBU | 8:110644774:T:C | −0.056 | 0.010 | $3.62\times10^{-9}$ | $2.04\times10^{-7}$ | 0.590 | 0.954 | 77.5% | 59.3% | 74.7% | 61.5% | 0 |
| rs1078514 | IRS2 | None | 13:110463168:T:C | 0.068 | 0.026 | $4.82\times10^{-3}$ | $2.30\times10^{-9}$ | $5.30\times10^{-3}$ | 0.261 | 33.3% | 39.2% | 40.6% | 10.7% | 0.99 |
| rs28648524 | USP8 | TRPM7 | 15:50787409:A:T | 0.064 | 0.010 | $6.45\times10^{-10}$ | $2.58\times10^{-8}$ | 0.406 | 0.661 | 78.1% | 53.7% | 76.5% | 79.8% | 1 |
| rs11650438 | PIGL | ADORA2B, ZSWIM7, PIGL, TTC19, NCOR1, CENPV, TRPV2 | 17:16233460:A:G | 0.050 | 0.009 | $2.93\times10^{-9}$ | $1.46\times10^{-7}$ | 0.528 | 0.288 | 46.9% | 17.8% | 48.5% | 64.0% | 0 |
| rs4485435 | FASN | None | 17:80045086:C:G | 0.082 | 0.014 | $2.61\times10^{-9}$ | N/A | N/A | N/A | 17.3% | 12.1% | 34.8% | 30.3% | 1 |
| rs6060983 | MYLK2 | None | 20:30420924:T:C | 0.069 | 0.037 | 0.0322 | $3.86\times10^{-9}$ | 0.035 | 0.149 | 69.3% | 99.0% | 71.8% | 29.0% | 0.23 |
| rs1736020 | USP25 | None | 21:16812552:A:C | 0.006 | 0.005 | 0.885 | $1.12\times10^{-9}$ | $4.74\times10^{-5}$ | 0.638 | 43.0% | 18.6% | 38.6% | 13.2% | 0.75 |
| rs73174657 | EP300 | ZC3H7B, POLR3H, CSDC2, PMM1, RANGAP1, MEI1, L3MBTL2, SLC25A17 | 22:41434158:A:G | −0.059 | 0.010 | $3.81\times10^{-9}$ | $4.90\times10^{-7}$ | 0.983 | 0.655 | 27.2% | 6.3% | 47.5% | 14.2% | 1 |
| rs10775809 | PPP6R2 | PPP6R2 | 22:50808017:A:T | 0.092 | 0.015 | $4.09\times10^{-10}$ | $5.61\times10^{-8}$ | 0.943 | 0.903 | 10.1% | 80.3% | 80.1% | 56.5% | 0.16 |

MR-MEGA could not be run for the lead SNP of the FASN locus, as it was missing in more than three cohorts: Foo et al.[1,2], 23andMe East Asian and 23andMe Latino. No P values were corrected for multiple tests. CHR, chromosome; BP, base pair; A1, effect allele; A2, other allele; BETA(RE), allelic effect in log odds ratio; SE, standard error; P(RE), two-sided P value of association from random effect; P(MR-MEGA): two-sided P value of association from MR-MEGA (chi-squared test with df = 4); P(ANC-HET), P value for the two-sided ancestral heterogeneity test (chi-squared test with df = 3); P(RES-HET): P value for the two-sided residual heterogeneity test (chi-squared test with df = 3); gnomAD [Ancestry] AF, A1 frequency reported for Europeans (EUR), East Asians (EAS), Amerindians (AMR) and Africans (AFR) by gnomAD v3.1.2; pLI, probability of being loss-of-function intolerant score from gnomAD v2.1.1 for the nearest coding gene (score was unavailable for gnomAD v3.1.2); SMR, summary-based Mendelian randomization; N/A, not available. Bolded are all significant P values (P<5×10⁻⁹ for the two-sided association tests, P<0.05 for the heterogeneity tests).

phagolysosome assembly, regulation of autophagosome maturation: $P_{FDR} < 0.03$) and tau protein (tau protein kinase activity: $P_{FDR} = 0.034$) were significant. At the tissue level, the genes of interest were enriched in all brain cell types, as well as pituitary tissue (Supplementary Fig. 9), consistent with the results from Nalls et al.[1].

When analyzing single-cell RNA-sequencing data, there was no expression enrichment across 88 brain cell types in mouse brain data when cross-referenced with DropViz[16] (Supplementary Fig. 10). There was also no enrichment of any specific cell types in the substantia nigra tissue in DropViz (Supplementary Fig. 10). However, in human midbrain data[17], dopaminergic (DA1) and GABAergic (GABA) neurons were enriched (Supplementary Fig. 10).

## eQTLs and SMR nominate 25 putative genes near novel loci

We also searched the GTEx v8 (ref. [18]) brain tissue eQTLs and multi-ancestry eQTL meta-analysis of the brain[19] to correlate novel loci with gene expression data (Supplementary Tables 10 and 11). To correlate potential putative genes with PD risk, we searched the significant-eQTL genes and genes near the loci with previously completed summary-based Mendelian randomization (SMR)[20] results in European-only data. When comparing the SNPs in novel loci with multi-ancestry brain eQTLs[19], 28 genes were significant (Supplementary Fig. 8 and Supplementary Tables 10 and 11). SMR found 25 genes in four novel loci associated with PD risk (Table 2 and Supplementary Table 12). Interestingly, *PPP6R2* and *CENPV* expression changes in substantia nigra were associated with PD risk. *PPP6R2* encodes protein phosphatase 6 regulatory subunit 2, a regulatory protein for protein phosphatase 6 catalytic subunit (*PPP6C*), which is involved in the vesicle-mediated transport pathway. Centromere protein V (*CENPV*) is involved in centromere formation and cell division.

## Discussion

This study is a large-scale GWAS meta-analysis of PD that incorporates multiple diverse ancestry populations. From the joint cohort analysis, we identified 66 independent risk loci near previously known PD risk regions and 12 potentially novel risk loci. Of the putative novel loci, nine had homogeneous effects and three had heterogeneous effects across the different cohorts. We found 17 additional suggestive loci using fixed-effects meta-analysis threshold at $P < 5\times10^{-8}$ and random-effects meta-analysis threshold at $P < 1\times10^{-6}$. We fine-mapped 23 loci by leveraging the diverse ancestry populations. We highlighted tissues and cell types associated with PD risk, which were consistent with previous findings[1]. Finally we used SMR to nominate 25 putative genes near our novel loci.

Novel loci contained genes in pathways previously implicated in PD. The *MTF2* and *PPP6R2* loci contain the genes *TMED5* and *PPP6R2*. Protein TMED5 localizes to Golgi body[21] and PPP6C, regulated by PPP6R2, is part of the vesicular transport pathways (https://reactome.org/content/detail/R-HSA-199977)[22], both of which are implicated in PD pathogenesis[23–28]. eQTL and SMR analysis showed association between expression changes for *PPP6R2* and *CENPV* in substantia nigra and PD risk. Because *substantia nigra* deterioration is a hallmark pathogenic feature of PD, *PPP6R2* and *CENPV* merit additional investigation. Within a known locus, a new independent signal was found in *RILPL2* (rs28659953). Protein RILPL2 interacts with LRRK2-phosphorylated Rab10 to block primary cilia generation[29]. Genes *JAK1* and *HS1BP3* are in two suggestive loci that were found only in Latin American and African populations. JAK1 is one of the proteins in the Janus kinase family, which is a critical part of the JAK-STAT pathway and is implicated in cytokine and inflammatory signaling[30]. *JAK1* variants have been implicated in autoimmune diseases such as juvenile idiopathic arthritis and multiple sclerosis[31]. *HS1BP3*, also known as essential tremor 2 (*ETM2*), has been implicated in essential tremor[32–34]. Based on its sequence, *ETM2* may modulate interleukin-2 signaling[35]. If these loci are confirmed, they would further support the growing appreciation for the role of

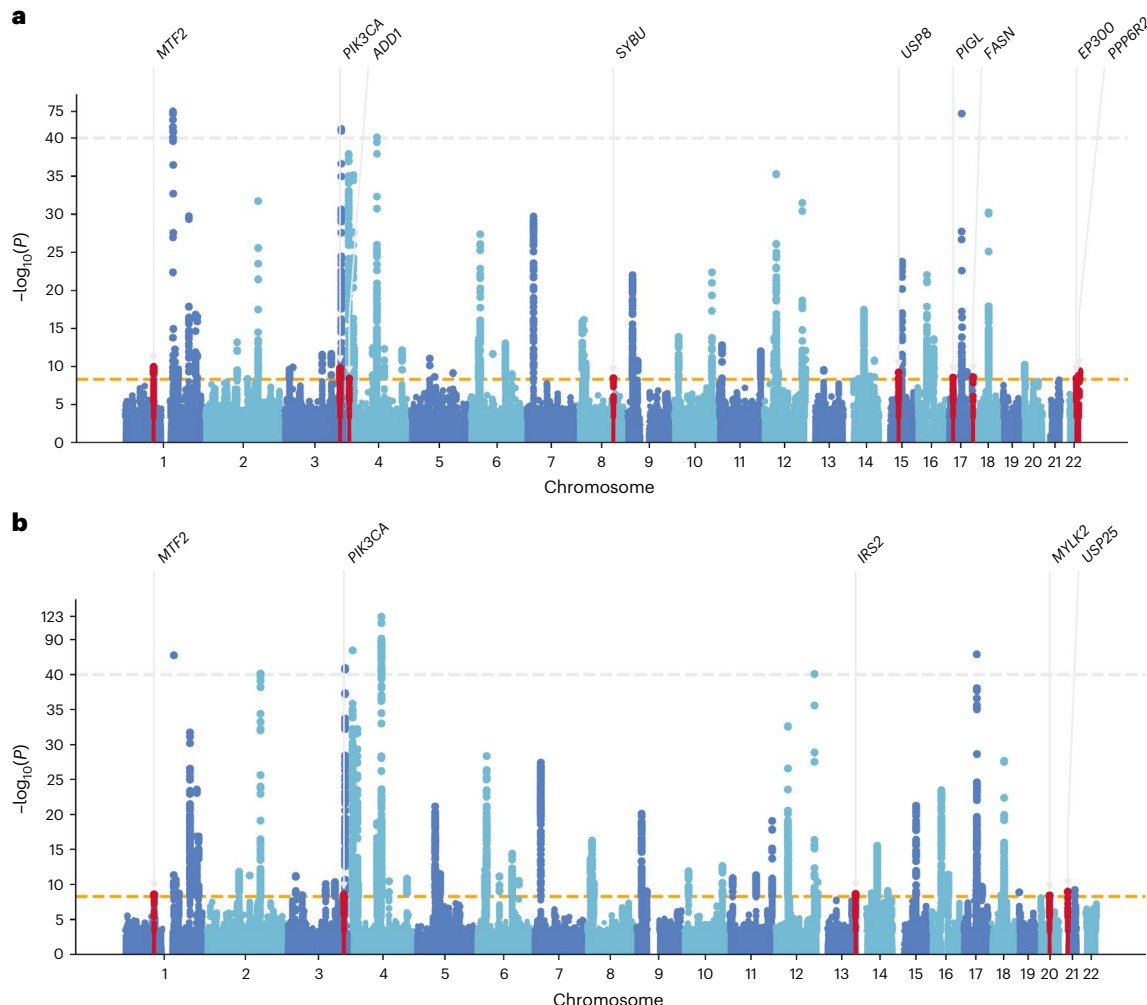

**Fig. 2 | Manhattan plots of the meta-analysis results across 2,525,730 participants. a**, Random-effects model test. **b**, MR-MEGA meta-regression test (chi-squared test with df = 4). The *x* axis shows chromosome and base pair positions of each variant tested in the meta-analyses. The *y* axis shows the two-sided *P* value with no multiple-test correction in the −log₁₀ scale. Orange horizontal dashed line indicates the Bonferroni-adjusted significant threshold of $P < 5 \times 10^{-9}$. Gray horizontal dashed line indicates the truncation line, where all −log₁₀ *P* values greater than 40 were truncated to 40 for visual clarity. Novel loci are highlighted in red and annotated with the nearest protein coding gene.

inflammation in PD[36]. All of the potentially novel PD loci identified in this analysis will require additional replication and functional validation to elucidate their role in PD pathogenesis. Previous findings in European populations found that polygenic risk scores explained 16–36% of PD heritability[1]. Although we did not perform similar tests incorporating our novel loci, they may explain additional heritable PD risk.

We found that 26 of the 66 detected known PD loci had nominally significant ancestral heterogeneity ($P_{\text{ANC-HET}} < 0.05$) and 10 remained significant after Bonferroni correction ($P_{\text{ANC-HET}} < 0.05/62$ MR-MEGA loci) (Fig. 3 and Supplementary Table 3). This heterogeneity may be caused by differences in effect sizes and allele frequencies between the different populations and thus should be studied as loci with potentially ancestrally divergent risk. 18 of the previous 92 known loci from single-ancestry GWASs did not overlap with any genome-wide significant loci in the multi-ancestry results at the significance threshold of $5 \times 10^{-9}$ (Supplementary Table 13). However, our results do not necessarily invalidate these previous results. First, several of the cohorts have small sample sizes, which may increase the influence of sampling variation. Another reason may be due to the stringent genome-wide significance threshold of $5 \times 10^{-9}$. Although this is a large PD GWAS meta-analysis, the more stringent significance threshold further raises

the sample size needed to achieve equivalent statistical power. Of the 17 European loci identified, 3 were significant at the $5 \times 10^{-8}$ threshold, and all 17 loci were at least nominally significant with the MR-MEGA method ($P_{\text{MR-MEGA}} < 5 \times 10^{-6}$). Lastly, variants may be more specific to the population in which they were first identified. 5 of the 17 variants had nominal ancestral heterogeneity ($P_{\text{ANC-HET}} < 0.05$). It is worth noting that there are large differences in statistical power across ancestries. Additional population-specific loci will likely reach significance when larger sample sizes are available for non-European datasets.

Our fine-mapping isolated several putative causal variants in previously discovered loci. *TMEM175*-rs34311866 has been previously identified as functionally relevant to PD risk[37], which is consistent with our fine-mapping results. Fine-mapped variants in *TMEM163*, *HIP1R* and *CAMK3D* were also found to be parts of active or strong transcription sites in substantia nigra tissues. Among the fine-mapped variants were two missense variants in *FCGR2A* and *SLC18B1*, albeit with a lower PP than the 7 singular putative variants that we highlighted in Table 3. *FCGR2A* is present in multiple immune-related ontology gene sets, further highlighting the potential role of the immune system in PD pathology. However, the function of *SLC18B1* is still unknown. Although the fine-mapping results provided by MR-MEGA are sufficient

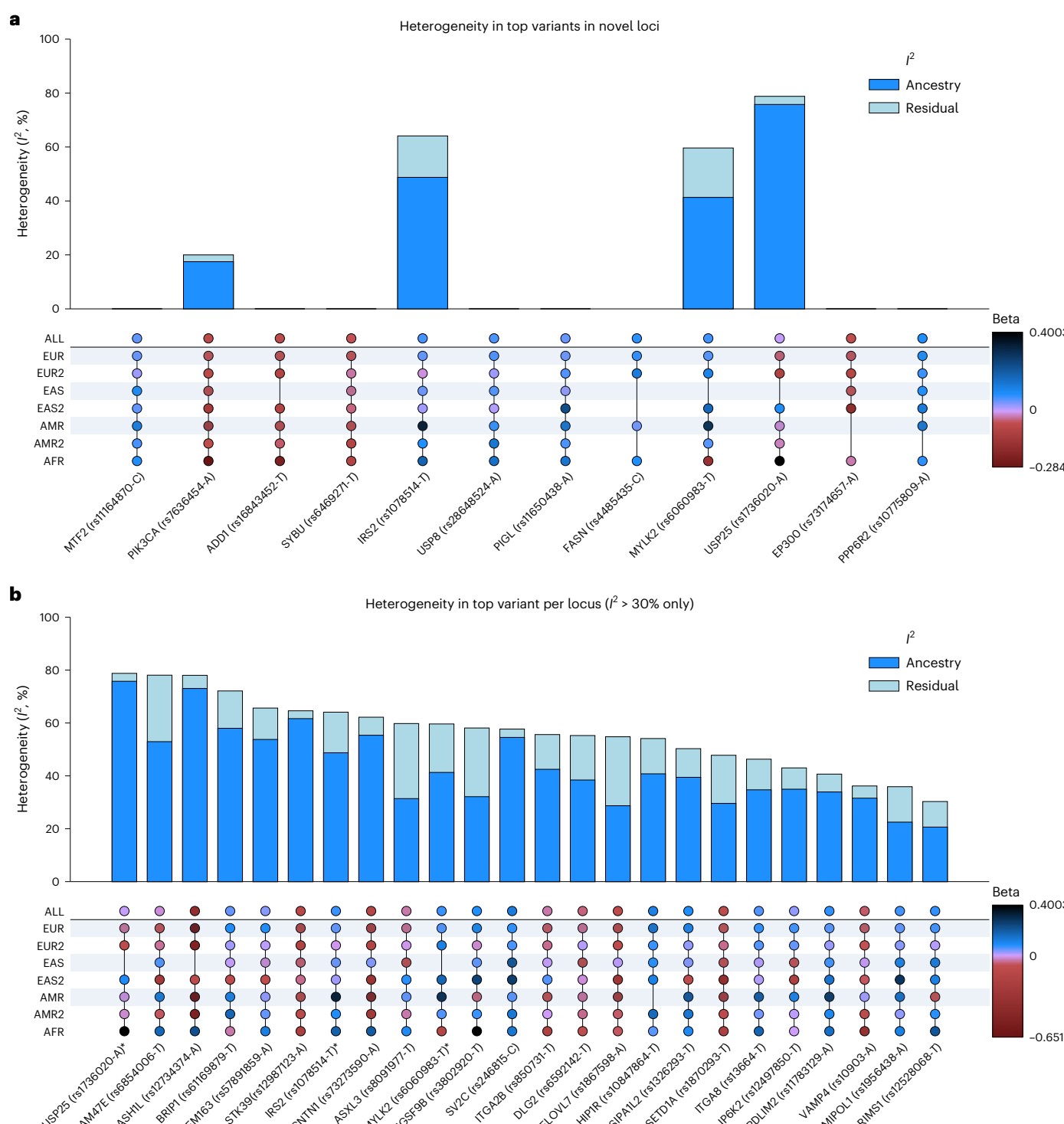

**Fig. 3 | Heterogeneity upset plots. a**, Top variants per novel loci. **b**, Top variants per MR-MEGA identified locus with moderate to high heterogeneity ($I^2 > 30$). The top bar plot illustrates heterogeneity with dark blue indicating ancestry heterogeneity proportion and light blue indicating other residual heterogeneity proportion. The bottom plot shows the subcohort level beta values with blue indicating positive and red indicating negative effect directions. Three variants with greater than 30% $I^2$ total heterogeneity were only identified in the MR-MEGA meta-analysis method, whereas little to no heterogeneity is observed in loci identified in random effect.

to identify putative causal variants for loci driven by one independent signal, multiple variants in a locus can contribute to complex traits. The additive and epistatic effects of multiple causal variants in a locus can be difficult to interpret when the effects associated with each independent signal are small.

The gene ontology analysis found multiple pathways that may be relevant to PD pathology (Supplementary Table 9), including those related to mitochondria (response to mitochondrial depolarization) vesicles (vesicle uncoating, phagolysosome assembly, regulation of autophagosome maturation) tau protein (tau protein kinase activity)

**Table 3 | MR-MEGA fine-mapping results for loci with a single SNP within the 95% credible set**

| Locus | Number of significant SNPs | Nominated variant | CHR:BP:A1:A2 | Nearest gene | Known PD gene ± 1 MB | Functional consequence | CADD | RDB |
|---|---|---|---|---|---|---|---|---|
| 11 | 6 | rs57891859 | 2:135464616:A:G | TMEM163 | TMEM163 | intronic | 6.746 | 4 |
| 19 | 926 | rs34311866 | 4:951947:C:T | TMEM175 | TMEM175 | exonic | 11.09 | NA |
| 23 | 1483 | rs356182 | 4:90626111:A:G | SNCA | SNCA | ncRNA intronic | 8.962 | NA |
| 24 | 121 | rs13117519 | 4:114369065:T:C | CAMK2D | CAMK2D | intergenic | 1.216 | 3a |
| 45 | 1371 | rs10847864 | 12:123326598:G:T | HIP1R | HIP1R | intronic | 2.403 | 2b |
| 60 | 1 | rs55818311 | 19:2341047:C:T | SPPL2B | LSM7 | ncRNA exonic | 1.096 | 5 |

Known PD genes are either known PD risk genes (SNCA and TMEM175) or genes with the highest score in the nearest known PD locus by the PD GWAS Locus Browser[37]. CHR, chromosome; BP, base pair; A1, effect allele; A2, other allele; CADD, combined annotation-dependent depletion score; RDB, regulomeDB score; ncRNA, non-coding RNA.

and immune cells (microglial cell/macrophage proliferation, and natural killer T cell differentiation)[36]. Neither mitochondrial nor immune cell pathways were significant in the previous European-only meta-analysis. Novel signals from the multi-ancestry approach may have given enough power to highlight these ontology terms. Out of 10 ontology terms that were significant in the previous European-only meta-analysis[1], 4 terms were not tested due to version differences in MSigDB and only 2 of the remaining terms were significant. However, the other 4 terms were still nominally significant at $P < 0.05$. This may be due to genome-wide signals that were less significant due to their heterogeneity across the different populations.

Although this is a large multi-ancestry PD meta-analysis GWAS, the European population is still overrepresented. Around 80% of full PD cases are of European descent. Individuals of African descent were particularly underrepresented at just 0.5% of the effective PD cases. The discoveries in our study warrant future efforts to expand studies in more diverse populations. The Global Parkinson's Genetics Program (GP2) is partnering with institutions that care for underrepresented populations to generate data for these underserved communities all over the world[5], and we will continue the ongoing analysis as more participants are genotyped. Just as the first PD GWASs failed to identify significant signals[38,39], we are confident that future diverse ancestry GWAS will produce impactful association results as sample sizes increase. Further efforts in multi-ancestry and non-European GWAS will identify loci that are more relevant to the global population and will continue to facilitate fine-mapping efforts to identify the genetic variants that drive these associations.

## Online content

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

## the 23andMe Research Team

**Stella Aslibekyan[9], Adam Auton[9], Elizabeth Babalola[9], Robert K. Bell[9], Jessica Bielenberg[9], Katarzyna Bryc[9], Emily Bullis[9], Paul Cannon[9], Daniella Coker[9], Gabriel Cuellar Partida[9], Devika Dhamija[9], Sayantan Das[9], Sarah L. Elson[9], Nicholas Eriksson[9], Teresa Filshtein[9], Alison Fitch[9], Kipper Fletez-Brant[9], Pierre Fontanillas[9], Will Freyman[9], Julie M. Granka[9], Karl Heilbron[9], Alejandro Hernandez[9], Barry Hicks[9], David A. Hinds[9], Ethan M. Jewett[9], Yunxuan Jiang[9], Katelyn Kukar[9], Alan Kwong[9], Keng-Han Lin[9], Bianca A. Llamas[9], Maya Lowe[9], Jey C. McCreight[9], Matthew H. McIntyre[9], Steven J. Micheletti[9], Meghan E. Moreno[9], Priyanka Nandakumar[9], Dominique T. Nguyen[9], Elizabeth S. Noblin[9], Jared O'Connell[9], Aaron A. Petrakovitz[9], G. David Poznik[9], Alexandra Reynoso[9], Madeleine Schloetter[9], Morgan Schumacher[9], Anjali J. Shastri[9], Janie F. Shelton[9], Jingchunzi Shi[9], Suyash Shringarpure[9], Qiaojuan Jane Su[9], Susana A. Tat[9], Christophe Toukam Tchakouté[9], Vinh Tran[9], Joyce Y. Tung[9], Xin Wang[9], Wei Wang[9], Catherine H. Weldon[9], Peter Wilton[9] & Corinna D. Wong[9]**

## the Global Parkinson's Genetics Program (GP2)

**Emilia M. Gatto[17], Marcelo Kauffman[18], Samson Khachatryan[19], Zaruhi Tavadyan[19], Claire E. Shepherd[20], Julie Hunter[21], Kishore Kumar[22], Melina Ellis[23], Miguel E. Rentería[24], Sulev Koks[25], Alexander Zimprich[26], Artur F. Schumacher-Schuh[27], Carlos Rieder[28], Paula Saffie Awad[29], Vitor Tumas[30], Sarah Camargos[31], Edward A. Fon[32], Oury Monchi[33], Ted Fon[34], Benjamin Pizarro Galleguillos[35], Marcelo Miranda[36], Maria Leonor Bustamante[37], Patricio Olguin[35], Pedro Chana[38], Beisha Tang[39], Huifang Shang[40], Jifeng Guo[41], Piu Chan[42], Wei Luo[43], Gonzalo Arboleda[44], Jorge Orozc[45], Marlene Jimenez del Rio[46], Alvaro Hernandez[47], Mohamed Salama[48], Walaa A. Kamel[49], Yared Z. Zewde[50], Alexis Brice[51], Jean-Christophe Corvol[52], Ana Westenberger[53], Anastasia Illarionova[54], Brit Mollenhauer[55], Christine Klein[53], Eva-Juliane Vollstedt[53], Franziska Hopfner[56], Günter Höglinger[56], Harutyun Madoev[53], Joanne Trinh[53], Johanna Junker[53], Katja Lohmann[53], Lara M. Lange[53,57], Manu Sharma[58], Sergiu Groppa[59], Thomas Gasser[58], Zih-Hua Fang[60], Albert Akpalu[61], Georgia Xiromerisiou[62], Georgios Hadjigorgiou[62], Ioannis Dagklis[63], Ioannis Tarnanas[64], Leonidas Stefanis[65], Maria Stamelou[66], Efthymios Dadiotis[62], Alex Medina[67], Germaine Hiu-Fai Chan[68], Nancy Ip[69], Nelson Yuk-Fai Cheung[68], Phillip Chan[69], Xiaopu Zhou[69], Asha Kishore[70], K. P. Divya[71], Pramod Pal[72], Prashanth Lingappa Kukkle[73], Roopa Rajan[74], Rupam Borgohain[75], Mehri Salari[76], Andrea Quattrone[77],**

Enza Maria Valente[78], Lucilla Parnetti[79], Micol Avenali[78], Tommaso Schirinzi[80], Manabu Funayama[81], Nobutaka Hattori[82], Tomotaka Shiraishi[83], Altynay Karimova[84], Gulnaz Kaishibayeva[84], Cholpon Shambetova[85], Rejko Krüger[86], Ai Huey Tan[87], Azlina Ahmad-Annuar[87], Mohamed Ibrahim Norlinah[88], Nor Azian Abdul Murad[89], Shahrul Azmin[90], Shen-Yang Lim[87], Wael Mohamed[91], Yi Wen Tay[87], Daniel Martinez-Ramirez[92], Mayela Rodriguez-Violante[93], Paula Reyes-Pérez[94], Bayasgalan Tserensodnom[95], Rajeev Ojha[96], Tim J. Anderson[97], Toni L. Pitcher[97], Arinola Sanyaolu[98], Njideka Okubadejo[98], Oluwadamilola Ojo[99], Jan O. Aasly[100], Lasse Pihlstrøm[101], Manuela Tan[101], Shoaib Ur-Rehman[102], Diego Veliz-Otani[5,6], Mario Cornejo-Olivas[103], Maria Leila Doquenia[104], Raymond Rosales[104], Angel Vinuela[105], Elena Iakovenko[106], Bashayer Al Mubarak[107], Muhammad Umair[108], Eng-King Tan[15], Michelle Mulan Lian[7,8,167], Jia Nee Foo[7,8,168], Ferzana Amod[109], Jonathan Carr[110], Soraya Bardien[110], Beomseok Jeon[111], Yun Joong Kim[112], Esther Cubo[113], Ignacio Alvarez[114], Janet Hoenicka[115], Katrin Beyer[116], Maria Teresa Periñan[117], Pau Pastor[118], Sarah El-Sadig[119], Kajsa Brolin[120], Christiane Zweier[121], Gerd Tinkhauser[122], Paul Krack[121], Chin-Hsien Lin[123], Hsiu-Chuan Wu[124], Pin-Jui Kung[125], Ruey-Meei Wu[123], Yihru Wu[124], Rim Amouri[126], Samia Ben Sassi[127], A. Nazl Başak[128], Gencer Genc[129], Özgür Öztop Çakmak[128], Sibel Ertan[128], Alastair J. Noyce[2], Alejandro Martínez-Carrasco[13], Anette Schrag[13], Anthony Schapira[13], Camille Carroll[130], Claire Bale[131], Donald Grosset[132], Eleanor J. Stafford[13], Henry Houlden[13], Huw R. Morris[13,14], John Hardy[13], Kin Ying Mok[13], Mie Rizig[13], Nicholas Wood[13], Nigel Williams[133], Olaitan Okunoye[13], Patrick Alfryn Lewis[134], Rauan Kaiyrzhanov[13], Rimona Weil[13], Seth Love[135], Simon Stott[136], Simona Jasaityte[13], Sumit Dey[2], Vida Obese[13], Alberto Espay[137], Alyssa O'Grady[138], Andrew B. Singleton[1,4], Andrew K. Sobering[139], Bernadette Siddiqi[138], Bradford Casey[138], Brian Fiske[138], Cabell Jonas[140], Carlos Cruchaga[141], Caroline B. Pantazis[4], Charisse Comart[138], Claire Wegel[142], Cornelis Blauwendraat[1,4,167], Dan Vitale[1,3,4,167], Deborah Hall[143], Dena Hernandez[1], Ejaz Shiamim[144], Ekemini Riley[145], Faraz Faghri[3,4], Geidy E. Serrano[146], Hampton Leonard[1,3,4], Hirotaka Iwaki[1,3,4], Honglei Chen[147], Ignacio F. Mata[16], Ignacio Juan Keller Sarmiento[148], Jared Williamson[144], Jonggeol Jeffrey Kim[1,2,167], Joseph Jankovic[149], Joshua Shulman[149,150], Justin C. Solle[138], Kaileigh Murphy[138], Karen Nuytemans[151], Karl Kieburtz[152], Katerina Markopoulou[153], Kenneth Marek[154], Kristin S. Levine[3,4], Lana M. Chahine[155], Laura Ibanez[156], Laurel Screven[4], Lauren Ruffrage[157], Lisa Shulman[158], Luca Marsili[137], Maggie Kuhl[138], Marissa Dean[157], Mary B. Makarious[1,13,14], Mathew Koretsky[1,4], Megan J. Puckelwartz[148], Miguel Inca-Martinez[16], Mike A. Nalls[1,3,4,168], Naomi Louie[138], Niccolò Emanuele Mencacci[148], Roger Albin[159], Roy Alcalay[160], Ruth Walker[161], Sara Bandres-Ciga[1,4], Sohini Chowdhury[138], Sonya Dumanis[162], Steven Lubbe[148], Tao Xie[163], Tatiana Foroud[164], Thomas Beach[146], Todd Sherer[138], Yeajin Song[3,4], Duan Nguyen[165], Toan Nguyen[165] & Masharip Atadzhanov[166]

[17]Sanatorio de la Trinidad Mitre- INEBA, Buenos Aires, Argentina. [18]Hospital JM Ramos Mejia, Buenos Aires, Argentina. [19]Somnus Neurology Clinic, Yerevan, Armenia. [20]Neuroscience Research Australia, Sydney, New South Wales, Australia. [21]ANZAC Research Institute, Concord, New South Wales, Australia. [22]Garvan Institute of Medical Research and Concord Repatriation General Hospital, Darlinghurst, New South Wales, Australia. [23]Concord Hospital, Concord, New South Wales, Australia. [24]QIMR Berghofer Medical Research Institute, Herston, Queensland, Australia. [25]Murdoch University, Perth, Western Australia, Australia. [26]Medical University Vienna Austria, Vienna, Austria. [27]Universidade Federal do Rio Grande do Sul / Hospital de Clínicas de Porto Alegre, Porto Alegre, Brazil. [28]Federal University of Health Sciences of Porto Alegre, Porto Alegre, Brazil. [29]Universidade Federal do Rio Grande do Sul, Porto Alegre, Brazil. [30]University of São Paulo, São Paulo, Brazil. [31]Universidade Federal de Minas Gerais, Belo Horizonte, Brazil. [32]Montreal Neurological Institute, Montreal, Quebec, Canada. [33]Institut universitaire de gériatrie de Montréal, Montreal, Quebec, Canada. [34]McGill University, Montreal, Quebec, Canada. [35]Universidad de Chile, Santiago, Chile. [36]Fundación Diagnosis, Santiago, Chile. [37]Faculty of Medicine Universidad de Chile, Santiago, Chile. [38]CETRAM, Santiago, Chile. [39]Central South University, Changsha, China. [40]West China Hospital Sichuan University, Chengdu, China. [41]Xiangya Hospital, Changsha, China. [42]Capital Medical University, Beijing, China. [43]Zhejiang University, Hangzhou, China. [44]Universidad Nacional de Colombia, Bogotá, Colombia. [45]Fundación Valle del Lili, Santiago De Cali, Colombia. [46]University of Antioquia, Medellin, Colombia. [47]University of Costa Rica, San Jose, Costa Rica. [48]The American University in Cairo, Cairo, Egypt. [49]Beni-Suef University, Beni Suef, Egypt. [50]Addis Ababa University, Addis Ababa, Ethiopia. [51]Paris Brain Institute, Paris, France. [52]Sorbonne Université, Paris, France. [53]University of Lübeck, Lübeck, Germany. [54]Deutsches Zentrum für Neurodegenerative Erkrankungen, Göttingen, Germany. [55]University Medical Center Göttingen, Göttingen, Germany. [56]Department of Neurology, University Hospital, LMU Munich, Munich, Germany. [57]University Medical Center Schleswig-Holstein, Lübeck, Germany. [58]University of Tubingen, Tübingen, Germany. [59]University of Mainz, Mainz, Germany. [60]The German Center for Neurodegenerative Diseases, Göttingen, Germany. [61]University of Ghana Medical School, Accra, Ghana. [62]University of Thessaly, Volos, Greece. [63]Aristotle University of Thessaloniki, Thessaloniki, Greece. [64]Ionian University, Corfu, Greece. [65]Biomedical research Foundation of the Academy of Athens, Athens, Greece. [66]Diagnostic and Therapeutic Centre HYGEIA Hospital, Marousi, Greece. [67]Hospital San Felipe, Tegucigalpa, Honduras. [68]Queen Elizabeth Hospital, Kowloon, Hong Kong. [69]The Hong Kong University of Science and Technology, Kowloon, Hong Kong. [70]Aster Medcity, Kochi, India. [71]Sree Chitra Tirunal Institute for Medical Sciences and Technology, Thiruvananthapuram, India. [72]National Institute of Mental Health & Neurosciences, Bengaluru, India. [73]Manipal Hospital, Delhi, India. [74]All India Institute of Medical Sciences, Delhi, India. [75]Nizam's Institute of Medical Sciences, Hyderabad, India. [76]Shahid Beheshti University of Medical Science, Tehran, Iran. [77]Magna Græcia University of Catanzaro, Catanzaro, Italy. [78]University of Pavia, Pavia, Italy. [79]University of Perugia, Perugia, Italy. [80]University of Rome Tor Vergata, Rome, Italy. [81]Juntendo University, Tokyo, Japan. [82]Faculty of Medicine, Juntendo University, Tokyo, Japan. [83]Jikei University School of Medicine, Tokyo, Japan. [84]Institute of Neurology and Neurorehabilitation, Almaty, Kazakhstan. [85]Kyrgyz State Medical Academy, Bishkek, Kyrgyzstan. [86]University of Luxembourg, Luxembourg, Luxembourg. [87]University of Malaya, Kuala Lumpur, Malaysia. [88]Universiti Kebangsaan Malaysia, Selangor, Malaysia. [89]UKM Medical Molecular Biology Institute, Kuala Lumpur, Malaysia. [90]Universiti Kebangsaan Malaysia Medical Centre, Kuala Lumpur, Malaysia. [91]International Islamic University, Kuala Lumpur, Malaysia. [92]Tecnologico de Monterrey, Monterrey, Mexico. [93]Instituto Nacional de Neurologia y Neurocirugia, Mexico City, Mexico. [94]Universidad Nacional Autónoma de México, Mexico City, Mexico. [95]Mongolian National University of Medical Sciences, Ulaanbaatar, Mongolia. [96]Tribhuvan University, Kirtipur, Nepal. [97]University of Otago, Dunedin, New Zealand. [98]University of Lagos, Lagos, Nigeria. [99]College of Medicine of the University of Lagos, Lagos, Nigeria. [100]Norwegian University of Science and Technology, Trondheim, Norway. [101]Oslo University Hospital, Oslo, Norway. [102]University of Science and Technology Bannu, Bannu, Pakistan. [103]Universidad Cientifica del Sur, Lima, Peru. [104]Metropolitan Medical Center, Manila, Philippines. [105]University of Puerto Rico, San Juan, Puerto Rico. [106]Research Center of Neurology, Moscow, Russia. [107]King Faisal Specialist Hospital and Research Center, Riyadh, Saudi Arabia. [108]King Abdullah International Medical Research Center, Jeddah, Saudi Arabia. [109]University of KwaZulu-Natal, Durban, South Africa. [110]Stellenbosch University, Stellenbosch, South Africa. [111]Seoul National University Hospital, Seoul, South Korea. [112]Yongin Severance Hospital, Seoul, South Korea. [113]Hospital Universitario Burgos, Burgos, Spain. [114]University Hospital Mutua Terrassa, Barcelona, Spain.

[115]Institut de Recerca Sant Joan de Deu, Barcelona, Spain. [116]Research Institute Germans Trias i Pujol, Barcelona, Spain. [117]Instituto de Biomedicina de Sevilla, Seville, Spain. [118]University Hospital Germans Trias i Pujol, Barcelona, Spain. [119]Faculty of medicine university of Khartoum, Khartoum, Sudan. [120]Lund University, Lund, Sweden. [121]Inselspital Bern, University of Bern, Bern, Switzerland. [122]University Hospital Bern, Bern, Switzerland. [123]National Taiwan University Hospital, Taipei City, Taiwan. [124]Chang Gung Memorial Hospital, Taoyuan City, Taiwan. [125]National Taiwan University, Taipei City, Taiwan. [126]National Institute Mongi Ben Hamida of Neurology, Tunis, Tunisia. [127]Mongi Ben Hmida National Institute of Neurology, Tunis, Tunisia. [128]Koç University, Istanbul, Turkey. [129]Şişli Etfal Training and Research Hospital, Istanbul, Turkey. [130]University of Plymouth, Plymouth, UK. [131]Parkinson's UK, London, UK. [132]University of Glasgow, Glasgow, UK. [133]Cardiff University, Cardiff, UK. [134]Royal Veterinary College University of London, London, UK. [135]University of Bristol, Bristol, UK. [136]Cure Parkinson's, London, UK. [137]University of Cincinnati, Cincinnati, OH, USA. [138]The Michael J. Fox Foundation for Parkinson's Research, New York, NY, USA. [139]Augusta University / University of Georgia Medical Partnership, Augusta, GA, USA. [140]Mid-Atlantic Permanente Medical Group, Bethesda, MD, USA. [141]Washington University, St. Louis, MO, USA. [142]Indiana University, Bloomington, IN, USA. [143]Rush University, Chicago, IL, USA. [144]Kaiser Permanente, Oakland, CA, USA. [145]Coalition for Aligning Science, Washington, WA, USA. [146]Banner Sun Health Research Institute, Sun City, AZ, USA. [147]Michigan State University, East Lansing, MI, USA. [148]Northwestern University, Evanston, IL, USA. [149]Baylor College of Medicine, Houston, TX, USA. [150]Texas Children's Hospital, Houston, TX, USA. [151]University of Miami Miller School of Medicine, Miami, FL, USA. [152]Beth Israel Deaconess Medical Center, Boston, MA, USA. [153]North Shore University Health System, Chicago, IL, USA. [154]Institute for Neurodegenerative Disorders, New Haven, CT, USA. [155]University of Pittsburgh, Pittsburgh, PA, USA. [156]Washington University, Saint Louis, MO, USA. [157]University of Alabama at Birmingham, Birmingham, AL, USA. [158]University of Maryland, Baltimore, MD, USA. [159]University of Michigan, Ann Arbor, MI, USA. [160]Columbia University, New York, NY, USA. [161]James J. Peters Veterans Affairs Medical Center, New York, NY, USA. [162]Aligning Science Across Parkinson's, Washington, WA, USA. [163]University of Chicago, Chicago, IL, USA. [164]Indiana University School of Medicine, Indianapolis, IN, USA. [165]Hue University, Huế, Vietnam. [166]University of Zambia, Lusaka, Zambia.

## Methods

### Study design and cohort descriptions

We used a single joint meta-analysis study design to maximize statistical power[40]. We used datasets representing four different ancestry groups: European, East Asian, Latin American and African. The meta-analysis included 49,049 PD cases, 18,618 PD proxy cases (participant with a parent with PD) and 2,458,063 neurologically normal controls (Table 1 and Supplementary Table 1). GWAS results of European[1], East Asian[2] and Latin American[3] populations were previously reported. African dataset as well as the additional Latin American and East Asian PD GWAS summary statistics were provided by 23andMe. The Finnish PD GWAS summary statistics was acquired from FinnGen Release 4 (G6_PARKINSON_EXMORE). For the FinnGen data, we chose the endpoint 'Parkinson's disease (more controls excluded)' (G6_PARKINSON_EXMORE), which excludes control participants with psychiatric diseases or neurological diseases. Although some FinnGen GWAS results also include UK Biobank participants, our FinnGen data did not include any UK Biobank participants.

### 23andMe diverse ancestry data

All self-reported PD cases and controls from 23andMe provided informed consent and participated in the research online, under a protocol approved by the external AAHRPP-accredited institutional review board (IRB), Ethical & Independent Review Services (E&I Review). Participants were included in the analysis on the basis of consent status as checked at the time data analyses were initiated. The name of the IRB at the time of the approval was Ethical & Independent Review Services. Ethical & Independent Review Services was recently acquired, and its new name as of July 2022 is Salus IRB (https://www.versiticlinicaltrials.org/salusirb). Samples were genotyped on one of five genotyping platforms: V1 and V2, which are variants of Illumina HumanHap550+ BeadChip; V3, Illumina OmniExpress+ BeadChip; V4, Illumina custom array that includes SNPs overlapping V2 and V3 chips; or V5, Illumina Infinium Global Screening Array. For inclusion, samples needed a minimal call rate of 98.5%. Genotyped samples were then phased using either Finch or Eagle2 (ref. 41) (RRID:SCR_015991) and imputed using Minimac3 (RRID:SCR_009292) and a reference panel of 1000 Genomes Phase III[42] (GRCh38) and UK10K data[43]. For this study, samples were classified as African, East Asian or Latino using a genotype-based pipeline[44] consisting of a support vector machine and a hidden Markov model, followed by a logistic classifier to differentiate Latinos from African Americans. Unrelated individuals were included in the analysis, as determined via identity-by-descent (IBD). Variants were tested for association with PD status using logistic regression, adjusting for age, sex, the first five principal components and genotyping platform. Reported $P$ values were from a likelihood ratio test.

### MAMA

We performed MAMA of GWAS results using MR-MEGA v0.2 (ref. 7) and PLINK 1.9 (RRID:SCR_001757). MR-MEGA performs a meta-regression by generating axes of genetic variation for each cohort, which are then used as covariates in the meta-analysis to account for differences in population structure. Although MR-MEGA was able to generate four principal components as axes of genetic variation, three principal components visibly separated the super population ancestries and explained 98% of the population variance (Supplementary Fig. 7). Therefore, we used three principal components to minimize overfitting. MR-MEGA has reduced power to detect associations for variants with homogeneous effects across populations. It is therefore recommended to run MR-MEGA alongside another meta-analysis method. PLINK 1.9 was used to perform random-effect meta-analysis to detect homogenous allelic effects.

Before the analysis, all datasets were harmonized to genome build hg19 using CrossMap[45] (RRID:SCR_001173) and Python 3.7. All variants were filtered by imputation score ($r^2 > 0.3$) and minor allele frequency ≥0.001. Only autosomal variants were kept in the final results as sex-chromosome data were not available for all ancestries. In total 20,590,839 variants met the inclusion criteria. However, MR-MEGA has a cohort-number requirement that varies based on the number of axes of variation. Therefore, 5,662,641 SNPs present in at least 6 of the 7 cohorts were analyzed in the MR-MEGA analysis. Bonferroni-adjusted alpha was set to a more stringent $5 \times 10^{-9}$ for all MAMAs to account for the larger number of haplotypes resulting from the ancestrally diverse datasets[8]. Genomic inflations were measured for all cohorts and the meta-analysis. Inflation for cohorts with large discrepancy between the case and control numbers was normalized to 1,000 cases and 1,000 controls. All inflation was nominal and below 1.02 (Supplementary Figs. 1–3 and Supplementary Table 1). No genomic control was applied prior to meta-analysis.

We identified genomic risk loci within our meta-analysis results using Functional Mapping and Annotation (FUMA) v1.3.8 (refs. 11,12). In brief, FUMA first identifies independent significant SNPs in the GWAS results by clumping all significant variants with the $r^2$ threshold <0.6, and then a locus is defined by merging LD blocks of all independent significant SNPs within 250 kb of each other. Start and end of a locus is defined by identifying SNPs in LD with the independent significant SNPs ($r^2 \geq 0.6$) and defining a region that encompasses all SNPs within the locus. Lead SNPs within a locus are determined by further clumping the independent significant variants within the genomic locus ($r^2 \geq 0.1$). The 1000 Genome reference panel with all ancestries was used to calculate the $r^2$.

To determine if any associated loci in the meta-analysis were not previously identified, all significant SNPs were compared to the 92 known PD risk variants found in the previous two major meta-analyses[1,2]. Two variants identified in the Latin American admixture population[3] could not be replicated, as the variants and their proxies were removed during quality control. If a genomic risk locus contained a significant hit in either population within 250 kb, then the locus was considered a known hit. Otherwise the locus was considered a novel hit. Forest plots and QQ plots were generated using python 3.7 with seaborn v0.11.2 and matplotlib v3.5.1. Manhattan plots were generated using gwaslab v3.3.11.

### Fine-mapping

Fine-mapping was performed using MR-MEGA[7], which approximates a single-SNP Bayes factor in favor of association. This is reported as the natural log of Bayes factor (lnBF) per SNP in the MR-MEGA meta-analysis summary statistics. SNPs were selected at meta-GWAS significance level ($P < 5 \times 10^{-9}$). PPs of driving the association signal at each locus were calculated from the Bayes factor as follows:

$$\pi_j = \frac{\Lambda_j}{\sum_{j=1}^{n} \Lambda_j},$$

where $\Lambda_j$ is the Bayes factor of the $j$th SNP within a locus with $n$ number of SNPs. Credible sets of fewer than 5 SNPs with sum PP ($\pi_j$) greater than 0.95 were accepted as putative causal variants. We excluded results located in the major histocompatibility complex region and the MAPT locus due to their complex LD structure.

### Estimation of population-specific or shared causal variants at associated loci

Proportion of population-specific and shared causal variants (PESCA v0.3)[10] was used to estimate whether causal variants at the loci identified in the meta-analysis were population-specific or shared between two populations. In brief, genome-wide heritability was estimated for the European and East Asian GWAS summary statistics using LD score regression[6,46]. Summary statistics of both populations were intersected with common variants with the 1000 Genome reference panels provided by PESCA, which have already been LD pruned ($R^2 > 0.95$) and

low-frequency SNPs removed (minor allele frequency < 0.05). The intersected variants were further split according to independent LD regions from the European and East Asian populations. The genome-wide prior probabilities of population-specific and shared causal variants were calculated using default parameters or as otherwise recommended by PESCA; then the results were used to calculate the PP for each variant. When the lead SNP was unavailable in the results, proxy variants ($R^2 > 0.8$) were used to approximate the PP for each variant for East Asian and European ancestry using R 4.2.0 and LDlinkR v1.1.2 (ref. [47]). Other cohorts were not included due to sample size constraints for this method.

### Functional annotation and GSEA

Functional annotation of the discovery results utilizing publicly available annotation data was done using FUMA v1.3.8 (refs. [11,12]). The summary statistics were annotated by ANNOVAR[48] (RRID:SCR_012821) through the FUMA platform. Our meta-analysis results were analyzed using MAGMA[13] (RRID:SCR_001757) to check for enrichment in gene ontology terms and gene expression data from tissues in GTEx v8 (ref. [18]). We tested 16,992 gene sets and gene ontology terms from MSigDB v7 (ref. [15]) as well as single-cell RNA-sequencing expression data from mouse brain samples in DropViz[16] and human ventral midbrain samples[17]. Test parameters were set to default. MAGMA gene analysis was run with a custom 1000 Genome reference panel that had a similar proportion of European, East Asian, Latin American and African participants as our main analysis. In short, we added all European participants and randomly selected participants from the East Asian, Latin American and African populations until the ancestry proportions of the reference panel were matching the effective sample size proportions of our study. The MAGMA gene analysis results were then analyzed using gene set analysis for ontology terms and gene-property analysis for tissue specificity. Results were adjusted for multiple tests using Benjamini–Hochberg FDR correction with the alpha of 0.05. The significant ontology terms were analyzed again in conditional analyses to identify and filter terms that share the same signals. Conditional analyses rerun the analyses with significant ontology terms as additional covariates. This can identify terms that lose significance when 'conditioned' on another, which may mean the terms share an underlying signal. When a term lost significance while the paired term retained nominal significance, the term that was no longer significant was discarded. When both terms lost significance, both were retained but highlighted with the comment that the pairs need to be interpreted together. Tissue level enrichment analysis was done using the pre-processed GTEx gene expression dataset provided by FUMA investigators. Single-cell expression enrichment analyses were performed by uploading the MAGMA gene analysis results to the FUMA cell-type analysis tool, which runs the MAGMA gene-property analysis with the chosen RNA-sequencing data. Additional pathway analyses of genes mapped by FUMA SNP2GENE were performed through GENE2FUNC with default parameters.

SNPs in the novel loci were searched in multi-ancestry brain eQTL meta-analysis results[19] (under Synapse ID syn23204884). We used a $P$-value cutoff of $10^{-6}$ as previously described[19]. eQTL and GWAS comparison plots were generated using LocusCompareR[49]. Multi-SNP SMR was used to test if DNA methylation and/or RNA expression of genes near the novel loci were associated with PD risk[20]. The nearest genes from the lead SNPs, significant genes in MAMA brain eQTL results and significant genes in GTEx v8 brain tissue were chosen for SMR. In total, 44 genes near the novel loci were searched in a list of previously completed PD SMR results from European-only GWAS meta-analysis (https://www.ukbiobank.ac.uk/learn-more-about-uk-biobank/news/nightingale-health-and-uk-biobank-announces-major-initiative-to-analyse-half-a-million-blood-samples-to-facilitate-global-medical-research)[18,20,50–56]. Only tissues in the central nervous system, digestive system and blood were used due to their relevance to PD pathology. Methylation

probes were annotated using the Bioconductor R package IlluminaHumanMethylation450kanno.ilmn12.hg19 v0.6.0 (https://bioconductor.org/packages/release/data/annotation/html/IlluminaHumanMethylation450kanno.ilmn12.hg19.html). The association signals were adjusted using FDR correction with the alpha of 0.05 and all signals with $P_{HEIDI} < 0.05$ were removed due to heterogeneity.

### Reporting summary

Further information on research design is available in the Nature Portfolio Reporting Summary linked to this article.

## Data availability

GWAS summary statistics for Foo et al.[2] and Loesch et al.[3] are available upon request to the respective authors. The UKBB genotype and phenotype data are available through the UKBB web portal https://www.ukbiobank.ac.uk/. FinnGen summary statistics are available through the FinnGen website https://www.finngen.fi/. GWAS summary statistics for 23andMe datasets (post-Chang and data included in Chang et al.[57] and Nalls et al.[58]) will be made available through 23andMe to qualified researchers under an agreement with 23andMe that protects the privacy of the 23andMe participants. Please visit research.23andme.com/collaborate/#publication for more information and to apply to access the data. An immediately accessible version of the multi-ancestry summary statistics is available on the Neurodegenerative Disease knowledge Portal (https://ndkp.hugeamp.org/) excluding Nalls et al.[58], 23andMe post-Chang et al.[57] and Web-Based Study of Parkinson's Disease (PDWBS) but including all analyzed SNPs. Same summary statistics are also available at AMP-PD (https://amp-pd.org/) under GP2 Tier 1 access and GWAS Catalog (https://www.ebi.ac.uk/gwas/) under accession code GCST90275127 (http://ftp.ebi.ac.uk/pub/databases/gwas/summary_statistics/GCST90275001-GCST90276000/GCST90275127/). After applying with 23andMe, the full summary statistics including all analyzed SNPs and samples in this GWAS meta-analysis will be accessible to the approved researcher(s). MSigDb is available at http://software.broadinstitute.org/gsea/msigdb. GTEx is available at https://gtexportal.org/home/. Multi-ancestry brain eQTL data from Zeng et al.[19] are available at https://hoffmg01.hpc.mssm.edu/brema/. eQTL/mQTL/caQTL data used for SMR outside of MetaBrain[50] and eQTLGen[52] are available at https://yanglab.westlake.edu.cn/software/smr/#DataResource. MetaBrain eQTL data are available at https://www.metabrain.nl/. eQTLGen data are available at https://www.eqtlgen.org/. pQTL data from Wingo et al.[54] are available upon request to the respective author. UK Biobank-Nightingale metabolomic data used for SMR are available at https://gwas.mrcieu.ac.uk/.

## Code availability

The analysis pipeline code is available on GP2 github: (https://github.com/GP2code/GP2-Multiancestry-metaGWAS) and deposited on Zenodo (https://doi.org/10.5281/zenodo.8045547)[59].

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

## Acknowledgements

This work was supported by the following grants and institutions: Intramural Research Program of the National Institutes of Health (NIH), National Institute on Aging (NIA), NIH, Department of Health and Human Services (A.B.S., C.B. and M.A.N.); National Institute of Neurological Disorders and Stroke (project numbers ZO1 AG000535 and ZIA AG000949 to A.B.S., C.B. and M.A.N.) (grant number R01NS112499 to I.M.); Parkinson's Foundation (Stanley Fahn Junior Faculty Award and an International Research Grants Program award to I.M.), Michael J Fox Foundation (to I.M. and A.J.N); Aligning Science Across Parkinson's Global Parkinson's Genetic Project (ASAP-GP2) (to I.M. and A.J.N); American Parkinson's Disease Association (to I.M.); National Medical Research Council Singapore (Open Fund Large Collaborative Grant MOH-000207 to E.-K.T.) (Open Fund Individual Research Grant MOH-000559 to J.N.F.); and Singapore Ministry of Education Academic Research Fund (Tier 2 MOE-T2EP30220-0005 and Tier 3 MOE-MOET32020-0004 to J.N.F.). Participation in this project was part of a competitive contract awarded to Data Tecnica International by the NIH to support open science research. This research has been conducted using the UK Biobank Resource under Application Number 33601. We want to acknowledge the participants and investigators of FinnGen study. We thank the research participants and employees of 23andMe. Data used in the preparation of this article were obtained from Global Parkinson's Genetics Program (GP2). GP2 is funded by the Aligning Science Against Parkinson's (ASAP) initiative and implemented by the Michael J. Fox Foundation for Parkinson's Research (https://gp2.org). For a complete list of GP2 members, see https://gp2.org. This work used the computational resources of the NIH HPC Biowulf cluster (http://hpc.nih.gov).

## Author contributions

A.B.S., C.B., M.A.N., I.M. and J.N.F. conceived the project. C.B., M.A.N., I.M. and J.N.F. designed and supervised the project. K.H., J.N.F. and I.M. provided data. J.J.K., D.V., D.V.-O. and M.M.L. performed the experiment. J.L. and C.W.S. assisted with data visualization. H.I., H.L., M.B.M., E.-K.T., S.B.-C. and A.J.N. advised on the project. J.J.K. wrote the manuscript with input from all authors.

## Competing interests

K.H. and members of the 23andMe Research Team are employed by and hold stock or stock options in 23andMe. M.A.N.'s participation in this project was part of a competitive contract awarded to Data Tecnica International by the NIH to support open science research; he also currently serves on the scientific advisory board for Clover Therapeutics and is an advisor to Neuron23. A.J.N. reports consultancy and personal fees from AstraZeneca, AbbVie, Profile, Roche, Biogen, UCB, Bial, Charco Neurotech, uMedeor, Alchemab and Britannia outside the submitted work. The other authors declare no competing interests.

## Additional information

**Correspondence and requests for materials** should be addressed to Jonggeol Jeffrey Kim, Cornelis Blauwendraat, Mike A. Nalls, Jia Nee Foo or Ignacio Mata.

# Reporting Summary

## Statistics

For all statistical analyses, confirm that the following items are present in the figure legend, table legend, main text, or Methods section.

| n/a | Confirmed | |
|---|---|---|
| ☐ | ☒ | The exact sample size (*n*) for each experimental group/condition, given as a discrete number and unit of measurement |
| ☐ | ☒ | A statement on whether measurements were taken from distinct samples or whether the same sample was measured repeatedly |
| ☐ | ☒ | The statistical test(s) used AND whether they are one- or two-sided *Only common tests should be described solely by name; describe more complex techniques in the Methods section.* |
| ☐ | ☒ | A description of all covariates tested |
| ☐ | ☒ | A description of any assumptions or corrections, such as tests of normality and adjustment for multiple comparisons |
| ☐ | ☒ | A full description of the statistical parameters including central tendency (e.g. means) or other basic estimates (e.g. regression coefficient) AND variation (e.g. standard deviation) or associated estimates of uncertainty (e.g. confidence intervals) |
| ☐ | ☒ | For null hypothesis testing, the test statistic (e.g. *F*, *t*, *r*) with confidence intervals, effect sizes, degrees of freedom and *P* value noted *Give P values as exact values whenever suitable.* |
| ☒ | ☐ | For Bayesian analysis, information on the choice of priors and Markov chain Monte Carlo settings |
| ☒ | ☐ | For hierarchical and complex designs, identification of the appropriate level for tests and full reporting of outcomes |
| ☐ | ☒ | Estimates of effect sizes (e.g. Cohen's *d*, Pearson's *r*), indicating how they were calculated |

*Our web collection on statistics for biologists contains articles on many of the points above.*

## Software and code

Policy information about availability of computer code

| Data collection | No software was used. |
|---|---|
| Data analysis | Data harmonization was done on Python 3.7 and CrossMap. Meta-analyses were done in PLINK 1.9 and Meta-Regression of Multi-Ethnic Genetic Association (MR-MEGA). Putative burden analysis was done using population-specific/shared causal variants (PESCA). Gene-ontology and tissue enrichment tests were done in Functional Mapping and Annotation (FUMA) and Multi-marker Analysis of GenoMic Annotation (MAGMA). Variant level annotation was done using ANNOVAR through FUMA. Plots and other miscellaneous analyses were done in Python or R. Analysis scripts are available on Github: https://github.com/GP2code/GP2-Multiancestry-metaGWAS and deposited deposited on Zenodo under doi:10.5281/zenodo.8045547.<br><br>Programs and their respective versions:<br>PLINK v1.9<br>MR-MEGA v0.2<br>FUMA v1.3.8<br>MAGMA v1.08<br>ANNOVAR-last updated on Dec 5 2016<br>PESCA v0.3<br>Python 3.7<br>R 4.2.0<br><br>R packages used:<br>LDlinkR v1.1.217 |

LocusCompareR

Python packages:
gwaslab v3.3.11
seaborn v0.11.2
matplotlib v3.5.1

For manuscripts utilizing custom algorithms or software that are central to the research but not yet described in published literature, software must be made available to editors and reviewers. We strongly encourage code deposition in a community repository (e.g. GitHub). See the Nature Portfolio guidelines for submitting code & software for further information.

# Data

Policy information about availability of data

All manuscripts must include a data availability statement. This statement should provide the following information, where applicable:

- Accession codes, unique identifiers, or web links for publicly available datasets
- A description of any restrictions on data availability
- For clinical datasets or third party data, please ensure that the statement adheres to our policy

GWAS summary statistics for Foo et al. 2020 and Loesch et al. 2020 are available upon request to the respective authors. The UKBB genotype and phenotype data are available through the UKBB web portal https://www.ukbiobank.ac.uk/. FinnGen summary statistics are available through the FinnGen website https://www.finngen.fi/. GWAS summary statistics for 23andMe datasets (post-Chang and data included in Chang et al. 2017 and Nalls et al. 2014) will be made available through 23andMe to qualified researchers under an agreement with 23andMe that protects the privacy of the 23andMe participants. Please visit research.23andme.com/collaborate/#publication for more information and to apply to access the data. An immediately accessible version of the multi-ancestry summary statistics is available on the Neurodegenerative Disease knowledge Portal (https://ndkp.hugeamp.org/) excluding Nalls et al. 2014, 23andMe post-Chang et al. 2017 and Web-Based Study of Parkinson's Disease (PDWBS) but including all analyzed SNPs. Same summary statistics are also available at AMP-PD (https://amp-pd.org/) under GP2 Tier 1 access and GWAS Catalog under accession code GCST90275127 (http://ftp.ebi.ac.uk/pub/databases/gwas/summary_statistics/GCST90275001-GCST90276000/GCST90275127/). After applying with 23andMe, the full summary statistics including all analyzed SNPs and samples in this GWAS meta-analysis will be accessible to the approved researcher(s). MSigDb is available at http://software.broadinstitute.org/gsea/msigdb/. GTEx is available at https://gtexportal.org/home/. Multi-ancestry brain eQTL data from Zeng et al. 2020 are available at https://hoffmg01.hpc.mssm.edu/brema/. eQTL/mQTL/caQTL data used for SMR outside of MetaBrain and eQTLGen are available at https://yanglab.westlake.edu.cn/software/smr/#DataResource. MetaBrain eQTL data are available at https://www.metabrain.nl/. eQTLGen data are available at https://www.eqtlgen.org/. pQTL data from Wingo et al. are available upon request to the respective author. UK Biobank-Nightingale metabolomic data used for SMR are available at https://gwas.mrcieu.ac.uk/.

# Research involving human participants, their data, or biological material

Policy information about studies with human participants or human data. See also policy information about sex, gender (identity/presentation), and sexual orientation and race, ethnicity and racism.

| | |
|---|---|
| Reporting on sex and gender | All data used in this analysis used biological sex as a covariate in their respective studies. |
| Reporting on race, ethnicity, or other socially relevant groupings | We used data from four ancestrally-distinct populations: European, East Asian, African, and Latin American. Ancestry was defined as continent of ancestral origin defined by an individual's genotype information. We did not use any socially constructed variable such as race and ethnicity as a proxy for ancestry. As we used summary-level data as-is, we did not make specific ancestry determinations as the respective studies made the determinations using their individual-level data. For 23andMe data, ancestry was determined using a genotype-based pipeline consisting of a support vector machine and a hidden Markov model, followed by a logistic classifier to differentiate Latinos from African Americans. |
| Population characteristics | Data included participants with Parkinson's disease and control participants. As we used data as-is, we do not describe any additional population characteristics and they can be found in their respective manuscripts |
| Recruitment | We did not recruit participants for this study. However, data from 23andMe participants were collected through self-report which may influence the results via self-report bias. |
| Ethics oversight | All self-reported PD cases and controls from 23andMe provided informed consent and participated in the research online, under a protocol approved by the external AAHRPP-accredited IRB, Ethical & Independent Review Services (E&I Review). Participants were included in the analysis on the basis of consent status as checked at the time data analyses were initiated. The name of the IRB at the time of the approval was Ethical & Independent Review Services. Ethical & Independent Review Services was recently acquired, and its new name as of July 2022 is Salus IRB (https://www.versiticlinicaltrials.org/salusirb). |

Note that full information on the approval of the study protocol must also be provided in the manuscript.

# Field-specific reporting

Please select the one below that is the best fit for your research. If you are not sure, read the appropriate sections before making your selection.

☒ Life sciences ☐ Behavioural & social sciences ☐ Ecological, evolutionary & environmental sciences

For a reference copy of the document with all sections, see nature.com/documents/nr-reporting-summary-flat.pdf

# Life sciences study design

All studies must disclose on these points even when the disclosure is negative.

| | |
|---|---|
| Sample size | We collected datasets from as many datasets as available. As a meta-analysis of 7 different studies/analyses, this work is the largest Parkinson's disease Genome-Wide meta-analysis to-date with 49049 cases, 18618 proxy-cases, and 2458063 controls. |
| Data exclusions | For 23andMe datasets, only data that met the quality control criteria and were unrelated were included in the analysis. For inclusion, samples needed a minimal call rate of 98.5%. Only data from unrelated participants were used to minimize bias from relatedness. We used data from other studies as-is, but each respective studies performed their own quality-control procedures, including removing related participants and low genotype quality samples. |
| Replication | We used a single joint meta-analysis study design to maximize statistical power. No replication samples were available as all available datasets were used in the meta-analysis. |
| Randomization | The experimental groups were divided into Parkinson's disease cases and controls. All data used were adjusted for age, sex, population structure-principal components to account for population stratification. |
| Blinding | Blinding is not relevant to study as this is a meta-analysis of observational genetic studies and not a randomized experiment. |

# Reporting for specific materials, systems and methods

We require information from authors about some types of materials, experimental systems and methods used in many studies. Here, indicate whether each material, system or method listed is relevant to your study. If you are not sure if a list item applies to your research, read the appropriate section before selecting a response.

## Materials & experimental systems

| n/a | Involved in the study |
|---|---|
| ☒ | ☐ Antibodies |
| ☒ | ☐ Eukaryotic cell lines |
| ☒ | ☐ Palaeontology and archaeology |
| ☒ | ☐ Animals and other organisms |
| ☒ | ☐ Clinical data |
| ☒ | ☐ Dual use research of concern |
| ☒ | ☐ Plants |

## Methods

| n/a | Involved in the study |
|---|---|
| ☒ | ☐ ChIP-seq |
| ☒ | ☐ Flow cytometry |
| ☒ | ☐ MRI-based neuroimaging |

