## [Peer Review File · Nature Genetics]

Peer Review Information

Manuscript Title: Multi-ancestry genome-wide association meta-analysis of Parkinson's disease

Corresponding author name(s): Mr Jonggeol (Jeffrey) Kim

Reviewer Comments & Decisions:

Decision Letter, initial version:
--

1st November 2022

Dear Jeff,

Your Analysis "Multi-ancestry genome-wide meta-analysis in Parkinson's disease" has been seen by two referees. You will see from their comments below that, while they find your work of potential interest, they have raised substantial concerns that must be addressed. In light of these comments, we cannot accept the manuscript for publication at this time, but we would be very interested in considering a suitably revised version that addresses the referees' concerns.

We hope you will find the referees' comments useful as you decide how to proceed. If you wish to submit a substantially revised manuscript, please bear in mind that we will be reluctant to approach the referees again in the absence of major revisions.

To guide the scope of the revisions, the editors discuss the referee reports in detail within the team, including with the chief editor, with a view to identifying key priorities that should be addressed in revision, and sometimes overruling referee requests that are deemed beyond the scope of the current study. In this case, we ask that you carefully address all technical queries related to the genome-wide association and downstream analyses, clarifying the presentation and interpretations where needed and extending the analyses where feasible as requested by the referees. We hope you will find this prioritized set of referee points to be useful when revising your study. Please do not hesitate to get in touch if you would like to discuss these issues further.

If you choose to revise your manuscript taking into account all reviewer and editor comments, please highlight all changes in the manuscript text file. At this stage, we will need you to upload a copy of the manuscript in MS Word .docx or similar editable format.

*2) If you have not done so already please begin to revise your manuscript so that it conforms to our Analysis format instructions, available [here](http://www.nature.com/ng/authors/article_types/index.html). Refer also to any guidelines provided in this letter.

[redacted]

If you wish to submit a suitably revised manuscript we would hope to receive it within 3-6 months. If you cannot send it within this time, please let us know. We will be happy to consider your revision so long as nothing similar has been accepted for publication at Nature Genetics or published elsewhere. Should your manuscript be substantially delayed without notifying us in advance and your article is eventually published, the received date would be that of the revised, not the original, version.

Thank you for the opportunity to review your work.

Sincerely,
Kyle

Kyle Vogan, PhD
Senior Editor
Nature Genetics
<https://orcid.org/0000-0001-9565-9665>

Referee expertise:

Referee #1: Genetics, neurodegenerative diseases

Referee #2: Genetics, neuroscience, statistical methods

Reviewers' Comments:

Reviewer #1:
Remarks to the Author:

In this article, Kim et al. report a first attempt at a meta-analysis of Parkinson's GWAS from multi-ancestry populations. The core of the analysis relies primarily on a tool called MR-MEGA, which was designed to specifically address multi-ethnic genetic associations. Beyond this specific point, the analyses are standard, and this is a well-conducted GWAS basic article (the term "basic" is not a criticism).

The QQ plots and genomic inflations generated for the meta-analyses are rather reassuring and the authors have been conservative by using a low significance level. However, one can wonder about the level of maturity of such an approach since 86% of the cases included in this study are of Caucasian origin (57,893 including Finns). Indeed, most of the other GWAS, even if presenting a large number of controls, ultimately bring very few cases (between 288 and 1633 for a total of 8967). At what level is the fact of finding 62 loci out of the 90 previously described expected? This is an important question since this study fully encompasses the GWASs of multi-ancestry European populations that were used to characterize these 90 loci. Does this mean that these unobserved signals are specific to European populations? Were these false positives generated in the previous study? Or are they false negatives (due to sampling variations related to the limited size of the other populations studied). It is obviously very difficult to answer this point, but this surely does not mean that the unobserved association of these 28 loci implies that these loci are incorrect. Indeed, it is in an involuntary way the message which could emerge from the reading of this article, and it seems to me it is essential that the authors pay attention to the use of the term "confirm" which should be avoided. In this case, this meta-analysis of GWAS data from multi-ancestry populations indicates that 62 loci are potentially involved in PD regardless of ethnic group. These results do not invalidate the involvement of the other 28 loci in Caucasian populations. This is a point that is really important because it can lead to confusion for non-specialists. A figure may be useful to better explain what the results means in terms genetic landscape between multi-ancestry populations.

A major problem is observed for the pathway analyzes between the authors' previous study and this one when the number of added cases is limited. How to explain that the number of significantly enriched gene sets increases from 11 to 111 and that, among the 11 initials, only 3 are in common. It is difficult to understand how this is possible except that there is significant heterogeneity in the global meta-analysis (beyond the characterized hits), which consequently impacts the MAGMA analysis (which is based on these global data and not just on GWAS-defined genes). The authors do not comment on this point, which nevertheless seems essential because it potentially indicates a problem inherent in the chosen approach. To be honest, I have no idea of the potential bias, but this difference is striking and deserves at least to be commented.

A minor point: Figure 1 is of poor quality and the legend should better describe the content of this figure. Under its current form, it is useless because it explains nothing for a non-specialist apart from a global strategy. Showing two subfigures, e.g. Random effects and MR-MEGA without explaining the interest of one over the other, is insufficient to hope that non-specialists will understand the interest of these approaches. Writers really need to be more educational.

Reviewer #2:

Remarks to the Author:

Parkinson's disease (PD) is a devastating disorder with a high individual and societal cost. Understanding the etiology of PD is of utmost importance in order to develop new therapeutics. In this study, Kim et al. performed a cross-ancestry meta-analysis of PD GWAS, followed by a number of post-GWAS analysis to identify risk genes and risk pathways. They identified 12 potential novel loci and fine-mapped 6 putative causal variants at 6 known PD loci. Unfortunately, this study reads like a first draft of what could be an important paper. The results are described superficially, the methods are not explained in sufficient details, and I have significant concerns regarding the integration of cross-ancestry summary stats with eQTL results discovered in another ancestry (as well as MAGMA/FUMA). I applaud the authors for doing this study but I believe that (a lot) more work is necessary for this study to reach the level of other recently published GWAS (e.g. Schwartzenuber et al. 2021, Bellenguez et al. 2022, Trubetsky et al. 2022).

Comments:

I don't have specific concerns regarding the meta-analysis and the authors appear to have used an appropriate method (MR-MEGA). However, the description of the results and the post-GWAS analysis are too superficial.

The authors report three loci with ancestral heterogeneity. It would be informative to investigate whether the three loci with ancestral heterogeneity are due to differences in allele frequencies across populations or not. In addition, did the authors investigate whether these loci were under selection?

The authors write that the IRS2 locus has the biggest departure in allelic effects in the Finnish cohort (Supplementary Figure 4). However, the index SNP is not described in the main text and the locus is not indicated in Sup. Fig. 4 (among many SNPs), which makes it difficult to find the IRS2 locus. The authors should indicate the loci on their supplementary figures so that the readers can easily connect

the text to the figures.

Authors should expand on the putative function of the protective variants (rs578139575 and rs73919910). These have very large effect sizes and could be extremely important. Are they coding/non-coding? What are the genes affected (closest gene if not coding)? What's their minor allele frequency? Any link between these genes and PD? Are the putative risk genes in PD relevant pathways?

I found the fine-mapping results very interesting but additional analysis would be interesting. Are the non-coding variants impacting transcription factor binding sites? Do they overlap epigenome marks in specific brain cell types (e.g. Nott et al. 2019, Corces et al. 2020)? If so, are these accessible regions connected to a gene (e.g. using ABC method, PLAC-seq from Nott et al. 2019)? Also, expanding on the loci with a small set of credible SNPs could be informative (e.g. number of loci with less than 5 SNPs in the 95% credible SNP set).

I have significant concerns regarding the use of FUMA with the LD structure of the full 1k genome dataset. The 1k genome dataset used could have a very different LD structure than the GWAS of the authors as the ancestry composition of the full 1k genome dataset is likely to be different from this GWAS. Most post-GWAS methods assume that the LD structure of the GWAS is similar to the LD structure of the reference panel and, so, it's difficult to evaluate whether the post-GWAS results are trustworthy or not. I suggest that the authors use the LD of their own study for FUMA/MAGMA (preferably); alternatively, they could sample the 1k genome data so that it better reflects the ancestry in their GWAS or perform FUMA/MAGMA for all ancestries separately (with the correct reference panel) and meta-analyze the gene-set enrichment results.

How do the authors explain the large increase in significant gene-sets using MAGMA? Could it be due to LD mismatches between the GWAS and the reference used for LD?

The authors do not describe their gene-set enrichment results which could be potentially interesting (e.g. lysosomal protein catabolic process, autophagy, etc.)? Conditional gene-set analysis could also help understand which gene sets are independently associated with PD as a lot are overlapping.

The cell-type enrichment was performed with a mouse single cell dataset from 2018. There are now much better human datasets (e.g. Siletti et al. 2022), and it would be more informative to use this instead of the mouse dataset.

The SMR analysis could be strengthened by running Coloc (Zuber et al. 2022). In addition, the authors did not filter their results based on the HEIDI test, resulting in potential false positives due to LD. More importantly, the eQTL studies are all based on European samples and cannot be used in SMR or Coloc with sumstats of a cross-ancestry GWAS. I suggest the authors to only run Coloc/SMR on the subset of their European samples.

More recent and more informative brain eQTL datasets should be used for the SMR/Coloc analysis. For example, the Metabrain study (de Klein et al. 2021), microglia eQTLs (Lopes et al. 2022, Kosoy et al. 2022), dopaminergic neurons eQTLs (Jerber et al. 2021), as well as eQTLs in multiple brain cell types (Bryois et al. 2022).

The chromosome X was not analyzed, it would be great to have it included.

20.5 million variants passed QC but only 5.6 million were used in the meta-analysis. It was not clear to me why those variants were lost. Can the authors find a way to use all their QC passed SNPs or better explain why it's necessary that they lose 75% of their SNPs?

There is a mention of PESCA in the methods section but it's unclear where it was used in the main text (if it was used).

Figures have a poor resolution. Please increase the resolution of all figures as I am unable to review them. This unfortunately gives an impression of sloppiness which decreases my confidence in the overall analysis (in addition to the PESCA methods not being linked apparently to anything in the text).

Why are there so many genes in sup. figure 5 and sup. figure 6? It looks like multiple genes are associated per locus.

Minor:

Was LDSC run to check that samples from different cohort did not overlap? It would also be interesting to see the genetic correlation between the different datasets (within ancestry).

Reporting the closest protein coding gene instead of the closest gene for the different loci would be more interesting to me as they are more likely to be functional.

Supplementary information mention 'PD MAMA' in multiple places. However, there is no mention of this in the main text. What does it refer to?

Table S10: why do some gene have an ensemble ID while others have a symbol? I suggest that the authors make two columns, one for the ensemble ID and one for the symbol.

Table S2: Authors should specify which is the effect allele (I guess A1).

Author Rebuttal to Initial comments

We would like to thank the editor for assisting with our submission process and the reviewers for spending their valuable time to review our manuscript. Our responses to the reviewers' comments are below. Reviewer comments are in blue. Our responses are in black.

Reviewer #1:

Remarks to the Author:

In this article, Kim et al. report a first attempt at a meta-analysis of Parkinson's GWAS from multi-ancestry populations. The core of the analysis relies primarily on a tool called MR-MEGA, which was designed to specifically address multi-ethnic genetic associations. Beyond this specific point, the analyses are standard, and this is a well-conducted GWAS basic article (the term "basic" is not a criticism).

The QQ plots and genomic inflations generated for the meta-analyses are rather reassuring and the authors have been conservative by using a low significance level. However, one can wonder about the level of maturity of such an approach since 86% of the cases included in this study are of Caucasian origin (57,893 including Finns). Indeed, most of the other GWAS, even if presenting a large number of controls, ultimately bring very few cases (between 288 and 1633 for a total of 8967). At what level is the fact of finding 62 loci out of the 90 previously described expected? This is an important question since this study fully encompasses the GWASs of multi-ancestry European populations that were used to characterize these 90 loci. Does this mean that these unobserved signals are specific to European populations? Were these false positives generated in the previous study? Or are they false negatives (due to sampling variations related to the limited size of the other populations studied). It is obviously very difficult to answer this point, but this surely does not mean that the unobserved association of these 28 loci implies that these loci are incorrect. Indeed, it is in an involuntary way the message which could emerge from the reading of this article, and it seems to me it is essential that the authors pay attention to the use of the term "confirm" which should be avoided. In this case, this meta-analysis of GWAS data from multi-ancestry populations indicates that 62 loci are potentially involved in PD regardless of ethnic group. These results do not invalidate the involvement of the other 28 loci in Caucasian populations. This is a point that is really important because it can lead to confusion for non-specialists. A figure may be useful to better explain what the results means in terms genetic landscape between multi-ancestry populations.

Thank you for reviewing our manuscript. You raise an important point about the number of Multi-Ancestry Meta-Analysis (MAMA) loci that overlapped with known loci in the European populations. A known locus may be a false negative in the MAMA for multiple reasons, whether

they be due to sampling variations from limited size of the smaller cohorts, the more conservative genome-wide significance threshold despite the cohort make-up of majority European cases, and a known locus being a more ancestry-specific signal. Indeed among the lead SNPs of the “rejected” known loci, 5 variants would pass the more common $P < 5 \times 10^{-8}$ threshold and 5 variants show significant ancestral heterogeneity ($P_{\text{ANC-HET}} < 0.05$). In addition 17 of the 18 loci were at least nominally significant with the MR-MEGA method ($P_{\text{MR-MEGA}} < 5 \times 10^{-6}$). We agree that the language we used to describe our “confirmation” of 62 of the 90 European risk loci can be confusing and misleading especially to non-specialists. We have edited the manuscript to clarify our message that these variants are not necessarily be previous false positives but may be false negatives due to multiple reasons:

“18 of the previous 92 known loci from single-ancestry GWASes did not overlap with any genome-wide significant loci in the multi-ancestry results at the significance threshold of 5×10^{-9} (Supplementary Table S4). However our results do not necessarily invalidate the previous results. First, several of the cohorts have small sample sizes, which raises the risk of sampling variation. Another reason may be due to the stringent genome-wide significance threshold of 5×10^{-9} . While this is the largest PD GWAS meta-analysis to date, the increased significance threshold further raises the number of sample sizes needed for statistical power. When looking at a lower significance threshold, 3 of the 17 loci identified in the European study were significant at the 5×10^{-8} threshold and all 17 loci were at least nominally significant with the MR-MEGA method ($P_{\text{MR-MEGA}} < 5 \times 10^{-6}$). Lastly, variants may be more specific to the population they were first identified with. 5 of these variants were found to have nominal ancestral heterogeneity ($P_{\text{ANC-HET}} < 0.05$). It is worth noting that there are large differences in statistical power for each included ancestry. Additional population-specific loci will likely reach significance when larger sample sizes are available for non- European datasets.”

While lower the significance threshold to the more common 5×10^{-8} would increase the statistical power and “recapture” 5 known loci, we felt that it was important to remain

conservative with our results and use the more stringent significance threshold to minimize any potential bias due to LD structure difference from the 14% of the non-European ancestry.

However we do recognize that our readers may be interested in the variants significant in the more common $P < 5 \times 10^{-8}$ threshold. To give a better understanding of the potential of the suggestive loci, we have now added additional discussion on two suggestive variants found in the more common $P_{FE} < 5 \times 10^{-8}$ and $P_{RE} < 1 \times 10^{-6}$ that have especially high effect sizes:

“Genes *JAK1* and *HS1BP3* are near two suggestive loci that were found only in Latin American and African populations. *JAK1* is one of the proteins in the Janus Kinase family, which is a critical part of the JAK-STAT pathway, implicated in

cytokine and inflammatory signaling²⁷. *JAK1* has been implicated in autoimmune diseases such as juvenile idiopathic arthritis and multiple sclerosis²⁷. *HS1BP3*, also known as Essential Tremor 2 (*ETM2*) has been implicated in essential tremor^{28–30}. Uniprot inferred from its sequence that it may modulate interleukin-2 signaling³¹. If these loci are confirmed, they would further support the growing role of inflammation in PD³².”

Finally we also updated figure 2 (see below) which now contains subfigure C and D which are heterogeneity upset plots that shows the quantified ancestral and residual heterogeneity in the novel variants (subplot C) and variants with moderate to high heterogeneity (subplot D). This hopefully addresses and clarifies some of the ancestral differences including why some signals are not detected in non-Europeans.

Figure 2: Manhattan plots and upset plot of PD MAMA results. A: random-effect; B: MR-MEGA. Dotted lines indicate the Bonferroni adjusted significant threshold of $P < 5 \times 10^{-9}$. All $-\log_{10}P$ values greater than 40 were truncated to 40 for visual clarity. Novel loci are highlighted in red and annotated by the nearest protein coding gene. C: Heterogeneity upset plot of the top hits in novel loci. The top bar plot illustrates heterogeneity with dark blue indicating ancestry heterogeneity proportion and light blue indicating other residual heterogeneity proportion. The bottom plot shows the subcohort level beta values with blue indicating positive and negative indicating negative effect directions. 3 variants with greater than 30% I^2 total heterogeneity were only identified in the MR-MEGA meta-analysis method, while little to no heterogeneity are observed in loci identified in random-effect. D: Heterogeneity upset plot of the top variant per MR-MEGA identified locus that had moderate to high heterogeneity ($I^2 > 30$). Variants in novel loci are annotated with *

A major problem is observed for the pathway analyzes between the authors' previous study and this one when the number of added cases is limited. How to explain that the number of significantly enriched gene sets increases from 11 to 111 and that, among the 11 initials, only

3 are in common. It is difficult to understand how this is possible except that there is significant heterogeneity in the global meta-analysis (beyond the characterized hits), which consequently impacts the MAGMA analysis (which is based on these global data and not just on GWAS- defined genes). The authors do not comment on this point, which nevertheless seems essential because it potentially indicates a problem inherent in the chosen approach. To be honest, I have no idea of the potential bias, but this difference is striking and deserves at least to be commented.

Thank you for noting this discrepancy in gene-ontology enrichment results between our MAMA results and the previous European-only results. Our analysis used the 1000 Genomes reference panel with all samples from all ancestries in the reference panel, which may have contributed to false signals. We have now re-run the MAGMA analysis for gene ontology, tissue specificity, and cell-type (single-cell) enrichment using a new reference panel that better represents the effective sample size proportion differences between the different ancestries (84.4% EUR, 11.1% EAS, 4.0% AMR, and 0.6% AFR). Indeed, our new MAGMA analysis highlights 43 different ontology terms, which is a large reduction from 111. Conditional analysis further removed 3 ontology terms, leaving us with 40 terms. The conditional analysis also paired 13 terms with at least one other term as sharing the same underlying signal, requiring these pairs to be interpreted together.

However this still leaves us with the discrepancy between the previous European-only ontology results and the new results. Some can be attributed to the different versions of gene sets, as we are using a newer version of MsigDB with significantly more ontology terms (16,992 terms in MAMA vs 10,651 terms in Nalls et al. 2019). In addition 5 of the 10 ontology terms significant in the previous European-only study are no longer present in the latest iteration of ontology terms in MsigDB and therefore not tested with the multi-ancestry results. Out of the 6 remaining terms, 2 terms were significantly associated in Multi-Ancestry Meta-Analysis after false-discovery-rate correction, while the other 4 were still nominally significant in the new results ($P < 0.05$). In addition the MAGMA test was run on the random-effect summary statistics, which penalizes the variants that displayed heterogeneity across the cohorts. It stands to reason that the gene sets that were less significant in the multi-ancestry results contained genes that have heterogeneous effects across the different cohorts. We have added these comments to the manuscript:

The gene ontology analysis found multiple pathways that may be relevant to PD pathology (Table S6). Multiple biological processes previously implicated in PD pathogenesis such as mitochondria (response to mitochondrial depolarization), vesicles (vesicle uncoating, phagolysosome assembly, regulation of autophagosome maturation), tau protein (tau protein kinase activity), and immune cells (microglial cell proliferation, macrophage proliferation, NK T Cell differentiation)³³. Neither mitochondria nor immune cell pathways were significant in the previous European-only meta-analysis. Novel signals from the multi-ancestry approach may have given enough power to highlight these ontology terms. Out of 10 ontology terms that were significant in the previous European-only meta-analysis¹, 4 terms were not tested due to version differences in MsigDB and only 2 of the remaining terms were significant after multiple test correction. However the other 4 terms were still nominally significant at $P < 0.05$. This may be due to genome-wide signals that were less significant due to their heterogeneity across the different populations.

A minor point: Figure 1 is of poor quality and the legend should better describe the content of this figure. Under its current form, it is useless because it explains nothing for a non-specialist apart from a global strategy. Showing two subfigures, e.g. Random effects and MR-MEGA without explaining the interest of one over the other, is insufficient to hope that non-specialists will understand the interest of these approaches. Writers really need to be more educational.

Thank you for the suggestions on how we can improve the figure 1. We have generated a more descriptive figure and a legend that better explains the difference between the two meta-analysis methods:

Figure 1 - Study design of Multi-Ancestry Meta-Analysis. Top: four ancestry groups used in the meta-analysis. Middle: the multi-ancestry meta-analysis and the two methods used. Random-effect is better

suited for risk variants with homogeneous effect direction across different ancestries while MR-MEGA can identify risk variants with heterogeneous effects due to population stratification introduced by ancestry differences. Bottom: downstream analyses and their examples.

Again, thank you for your comments and suggestions. They have greatly improved the manuscript.

Reviewer #2:

Remarks to the Author:

Parkinson's disease (PD) is a devastating disorder with a high individual and societal cost. Understanding the etiology of PD is of utmost importance in order to develop new therapeutics. In this study, Kim et al. performed a cross-ancestry meta-analysis of PD GWAS, followed by a number of post-GWAS analysis to identify risk genes and risk pathways. They identified 12 potential novel loci and fine-mapped 6 putative causal variants at 6 known PD loci. Unfortunately, this study reads like a first draft of what could be an important paper. The results are described superficially, the methods are not explained in sufficient details, and I have significant concerns regarding the integration of cross-ancestry summary stats with eQTL results discovered in another ancestry (as well as MAGMA/FUMA). I applaud the authors for doing this study but I believe that (a lot) more work is necessary for this study to reach the level of other recently published GWAS (e.g. Schwartzentruber et al. 2021, Bellenguez et al. 2022, Trubetskoy et al. 2022).

Thank you for reviewing our manuscript and pointing out the importance of this study. We also want to apologize for leaving out some of the details in our manuscript. We have made a significant effort to include additional language and analyses to better supplement our findings which we will describe further below.

Comments:

I don't have specific concerns regarding the meta-analysis and the authors appear to have used an appropriate method (MR-MEGA). However, the description of the results and the

post- GWAS analysis are too superficial.

The authors report three loci with ancestral heterogeneity. It would be informative to investigate whether the three loci with ancestral heterogeneity are due to differences in allele frequencies across populations or not. In addition, did the authors investigate whether these loci were under selection?

Thank you for this comment. Thanks to your feedback, we have added more details to our results and post-GWAS analyses, and we hope that this will be evident as we discuss these changes below. We have added ancestry-specific allele frequencies reported in gnomAD to Table 2:

Table 2. Meta-analysis results of lead SNPs in the novel loci. All significant P values are highlighted ($P < 5 \times 10^{-9}$ for the association tests, $P < 0.05$ for the heterogeneity tests). MR-MEGA could not be run for the lead SNP of the *FASN* locus as it was missing in more than 3 cohorts: Foo et al. 2020, LARGE-PD, and 23andMe Latino. A1: Effect Allele. A2: Other Allele. BETA(RE): allelic effect in log odds ratio. SE: standard error. P(RE): P value of association from Random-Effect. P(MR-MEGA): P value of association from MR-MEGA. P(ANC-HET): P value for the ancestral heterogeneity test. P(RES-HET): P value for the residual heterogeneity test. gnomAD [Ancestry] AF: A1 frequency reported by gnomAD v3.1.2. pLI: probability of being loss-of-function intolerant score from gnomAD v2.1.1 for the nearest coding gene (score was unavailable for gnomAD v3.1.2)

rsID	Nearest Coding Gene	CHR:BP:A1:A2	BETA(RE)	SE	P(RE)	P(MR-MEGA)	P(ANC-HET)	P(RES-HET)	gnomAD EUR AF	gnomAD EAS AF
rs11164870	MTF2	1:93552187:C:G	0.054	0.009	1.15E-10	2.64E-09	0.229	0.928	0.3902	0.3
rs6806917	PIK3CA	3:178861417:T:C	-0.070	0.011	1.65E-10	3.43E-09	0.215	0.762	0.8203	0.8
rs16843452	ADD1	4:2849168:T:C	-0.068	0.012	4.11E-09	3.19E-07	0.747	0.687	0.1845	0.4
rs6469271	SYBU	8:110644774:T:C	-0.056	0.010	3.62E-09	2.04E-07	0.590	0.954	0.7749	0.5
rs1078514	IRS2	13:110463168:T:C	0.068	0.026	0.004817	2.30E-09	5.30E-03	0.261	0.3328	0.3
rs28648524	USP8	15:50787409:A:T	0.064	0.010	6.45E-10	2.58E-08	0.406	0.661	0.7813	0.5

rs11650438	PIGL	17:16234260:A:G	0.050	0.009	2.93E-09	1.46E-07	0.528	0.288	0.4694	0.1
rs4485435	FASN	17:80045086:C:G	0.082	0.014	2.61E-09	N/A	N/A	N/A	0.1726	0.1
rs6060983	MYLK2	20:30420924:T:C	0.069	0.037	0.03221	3.86E-09	0.035	0.149	0.6926	0.9
rs1736020	USP25	21:16812552:A:C	0.006	0.005	0.8853	1.12E-09	4.74E-05	0.638	0.4297	0.1
rs73174657	EP300	22:41434158:A:G	-0.059	0.010	3.81E-09	4.90E-07	0.983	0.655	0.2723	0.06
rs10775809	PPP6R2	22:50808017:A:T	0.092	0.015	4.09E-10	5.61E-08	0.943	0.903	0.1010	0.8

The added gnomAD frequencies show a large range of frequencies across the different ancestry groups for the three novel loci with significant ancestral heterogeneity (AF ranges: rs1078514 = 0.1072 - 0.4056, rs6060983 = 0.2900 - 0.9895, rs1736020 = 0.1318 - 0.4297). Note that there are also significant allele frequency differences in variants identified by random-effects such as the *PIGL* locus, which has allele frequency ranging from 0.6402 in AFR to 0.1783 in EAS. In these situations the effect estimates may be similar across populations or these effect estimates may not be correlated with major ancestral dimensions. To check if the loci were under selection, we also added the probability of being loss-of-function intolerant (pLI) scores found in gnomAD for each nearest protein coding gene. We found that most genes were likely highly selected for as 7 out of 12 had the pLI score of 0.99 or 1. Genes with low pLI scores were found in both loci with no ancestry heterogeneity (*SYBU*, *PIGL*, *PPP6R2*) and significant ancestry heterogeneity (*MYLK2*). We added a brief description of the pLI score results in the manuscript:

“When looking at the nearest protein coding genes and their probability of being loss-of-function intolerant (pLI) score, we found that 7 out of 12 genes had the pLI score of 0.99 or 1. Genes with low pLI scores were found in both loci with no ancestry heterogeneity (*SYBU*, *PIGL*, *PPP6R2*) and significant ancestry heterogeneity (*MYLK2*).”

In variants with high heterogeneity, the ancestry and residual heterogeneity can be segregated with MR-MEGA, which can segregate the overall heterogeneity to ancestry and residual heterogeneity by incorporating axes of genetic variation from population allele frequencies. To make this clear, we have made two changes to the manuscript. First, we have added language in the results section to better explain the heterogeneity tests in MR-MEGA:

“The random-effects model has greater power to detect homogenous allelic effects⁶. MR-MEGA uses axes of genetic variation as covariates in its meta-regression analysis and has the greatest power to detect heterogeneous effects across the different cohorts. MR-MEGA also distinguishes ancestral heterogeneity (differences in effect estimates due to ancestry-level genetic variations) from residual heterogeneity using axes of genetic variation generated from the allele frequencies across the different cohorts.”

Second, we have revised Figure 2 to contain the subfigure C and D which are heterogeneity upset plots that shows the quantified ancestral and residual heterogeneity in the novel hits (C) and in hits with moderate to high heterogeneity ($I^2 > 30$). Close inspection reveals that the novel variants identified by random-effects have no heterogeneity, while those only identified by MR-MEGA have significant levels of ancestral heterogeneity:

Figure 2: Manhattan plots and upset plot of PD MAMA results. A: random-effect; B: MR-MEGA. Dotted lines indicate the Bonferroni adjusted significant threshold of $P < 5 \times 10^{-9}$. All $-\log_{10}P$ values greater than 40 were truncated to 40 for visual clarity. Novel loci are highlighted in red and annotated by the nearest protein coding gene. C: Heterogeneity upset plot of the top hits in novel loci. The top bar plot illustrates heterogeneity with dark blue indicating ancestry heterogeneity proportion and light blue indicating other residual heterogeneity proportion. The bottom plot shows the subcohort level beta values with blue indicating positive and negative indicating negative effect directions. 3 variants with greater than 30% I^2 total heterogeneity were only identified in the MR-MEGA meta-analysis method, while little to no heterogeneity are observed in loci identified in random-effect. D: Heterogeneity upset plot of the top variant per MR-MEGA identified locus that had moderate to high heterogeneity ($I^2 > 30$). Variants in novel loci are annotated with *

The authors write that the IRS2 locus has the biggest departure in allelic effects in the Finnish cohort (Supplementary Figure 4). However, the index SNP is not described in the main text and the locus is not indicated in Sup. Fig. 4 (among many SNPs), which makes it difficult to

find the *IRS2* locus. The authors should indicate the loci on their supplementary figures so that the readers can easily connect the text to the figures.

Thank you for your suggestion. We have edited the manuscript so that the manuscript uses the nearest protein coding gene when referring to a specific locus and the rsID when referring to the lead SNP. and For example:

“Lastly, the *USP25* locus had the most significant ancestral heterogeneity (lead SNP rs1736020, $P_{\text{ANC-HET}} = 4.74 \times 10^{-5}$)”

In addition we have added the locus number, nearest protein coding gene, and/or rsID on the supplementary figures. For example, this is the updated supplementary figure 4 for *IRS2* locus:

Forest Plot of MR-MEGA Locus 47 (rs1078514, *IRS2*)

Authors should expand on the putative function of the protective variants (rs578139575 and

rs73919910). These have very large effect sizes and could be extremely important. Are they coding/non-coding? What are the genes affected (closest gene if not coding)? What's their minor allele frequency? Any link between these genes and PD? Are the putative risk genes in PD relevant pathways?

Thank you for this comment. We have incorporated your suggestions and added additional language further describing these regions:

(In Results:) "Two loci near *JAK1* and *HS1BP3* were exclusively found in the 23andMe Latin American and African cohorts. The lead SNPs (rs578139575 and rs73919910) for these loci are non-coding and very rare in European populations but more common in Africans and Latin Americans (gnomAD v3.1.2 allele frequencies in EUR: 0.0001616, 0.002307; AFR: 0.01637, 0.08837; AMR:

0.004063, 0.01905). If confirmed, these loci would confer a large amount of PD protective effect (beta: -1.3, -0.54). These loci merit further studies in the African and Latino populations."

(In Discussions:) "Genes *JAK1* and *HS1BP3* are near two suggestive loci that were found only in Latin American and African populations. *JAK1* is one of the proteins in the Janus Kinase family, which is a critical part of the JAK-STAT pathway, implicated in cytokine and inflammatory signaling²⁷. *JAK1* has been implicated in autoimmune diseases such as juvenile idiopathic arthritis and multiple sclerosis²⁷. *HS1BP3*, also known as Essential Tremor 2 (*ETM2*) has been implicated in essential tremor²⁸⁻³⁰. Uniprot inferred from its sequence that it may modulate interleukin-2 signaling³¹. If these loci are confirmed, they would further support the growing role of inflammation in PD³²."

I found the fine-mapping results very interesting but additional analysis would be interesting. Are the non-coding variants impacting transcription factor binding sites? Do they overlap epigenome marks in specific brain cell types (e.g. Nott et al. 2019, Corces et al. 2020)? If so, are these accessible regions connected to a gene (e.g. using ABC method, PLAC-seq from Nott et al. 2019)? Also, expanding on the loci with a small set of credible SNPs could be informative (e.g. number of loci with less than 5 SNPs in the 95% credible SNP set).

Thank you for your suggestion. We agree with your comment and we have now modified our credible SNP set threshold to 95% posterior probability (PP) and generated an annotated list of SNPs for loci with fewer than 5 SNPs within the credible SNP set. As for PLAC-seq and other methods that look at overlaps with epigenome marks in specific brain cell types, close external collaborators are doing similar work and out of respect we did not run such analyses. However, we have investigated variants with PP > 95% in RegulomeDB, which found several to be present in active transcription start sites in *substantia nigra* and astrocytes. We have added and discussed these results:

“Fine-mapping was also performed using MR-MEGA, which uses ancestry heterogeneity to increase resolution. We identified 25 loci that had fewer than 5 variants within the 95% credible set. Of these, MR-MEGA nominated a single putative causal variant with > 95% PP in 6 loci: *TMEM163*, *TMEM175*, *SNCA*, *CAMK2D*, *HIP1R*, and *LSM7* (Table 4). Our results affirmed previous results showing the *TMEM175* p.M393T coding variant as the likely causal variant¹⁰. The putative variants near *LSM7* and *HIP1R* have strong evidence for regulome binding (RegulomeDB rank ≥ 2). In particular the *HIP1R* variant rs10847864 is located in active transcription start site in *substantia nigra tissue* (chromatin state windows: chr12:123326200..123327200) and astrocytes in the spinal cord and the brain (chromatin state windows: chr12:123326400..123326600). Outside of the singular putative sets, we identified missense variants in 2 genes: *FCGR2A* (p.H167R, PP = 0.145) and *SLC18B1* (p.S30P, PP = 0.780).”

And extended our discussion of the fine-mapped variants:

“Our fine mapping isolated several putative causal variants in previously discovered loci. *TMEM175*-rs3431866 has been previously identified as functionally relevant to PD risk¹⁰, which is consistent with our fine-mapping results. Fine-mapped variant in *HIP1R* was found to be part of active transcription start sites in *substantia nigra* tissues and astrocytes. Among the fine-mapped variants were 2 missense variants in *FCGR2A* and *SLC18B1*, albeit at lower PP than the 7 putative causal variants we highlighted in Table 3.

FCGR2A is present in immune-related ontology terms, further highlighting the potential role of the immune system in PD pathology. However the function of *SLC18B1* is still unknown. While the fine-mapping results provided by MR-MEGA are sufficient to identify putative causal variants for loci driven by one independent signal, multiple variants in a locus can contribute to complex traits. The additive and epistatic effects of multiple causal variants in a locus can be difficult to interpret when the effects associated with each independent signal are small.”

We have also removed *MAPT* from our results as it has a complex LD pattern that makes it difficult to fine-map using statistical methods. We have noted this in the methods: “We excluded results located in the major histocompatibility complex (MHC) region and the *MAPT* locus due to their complex LD structures.”

I have significant concerns regarding the use of FUMA with the LD structure of the full 1k genome dataset. The 1k genome dataset used could have a very different LD structure than the GWAS of the authors as the ancestry composition of the full 1k genome dataset is likely to be different from this GWAS. Most post-GWAS methods assume that the LD structure of the GWAS is similar to the LD structure of the reference panel and, so, it’s difficult to evaluate whether the post-GWAS results are trustworthy or not. I suggest that the authors use the LD of their own study for FUMA/MAGMA (preferably); alternatively, they could sample the 1k genome data so

that it better reflects the ancestry in their GWAS or perform FUMA/MAGMA for all ancestries separately (with the correct reference panel) and meta-analyze the gene-set enrichment results.

How do the authors explain the large increase in significant gene-sets using MAGMA? Could it be due to LD mismatches between the GWAS and the reference used for LD?

The authors do not describe their gene-set enrichment results which could be potentially interesting (e.g. lysosomal protein catabolic process, autophagy, etc.)? Conditional gene-set analysis could also help understand which gene sets are independently associated with PD as a lot are overlapping.

We share your concern regarding the potential differences in the LD structure of the 1000 Genome dataset and our cohort. As recommended, we have rerun the MAGMA analysis with a custom 1000 Genome dataset with ancestry proportions that better reflect the effective sample size proportions of our study (84.4% EUR, 11.1% EAS, 4.0% AMR, and 0.6% AFR). This better reflected the LD structure of our study and significantly reduced the number of significant gene ontology terms to 43. The conditional analysis further removed 3 ontology terms leaving us with 40 terms. The conditional analysis also paired 13 terms with at least one other term as sharing the same underlying signal, requiring these pairs to be interpreted together.

Nevertheless 40 ontology terms are still greater than 10 found in the European-only study and some of this may be explained by the different versions of MsigDB between the previous European-only analysis and our multi-ancestry analysis. In the multi-ancestry analysis, we have tested 16,992 gene sets while the previous European meta-analysis tested 10,652 gene sets.

This nearly 60% increase in number ontology terms in addition to increase in sample size may contribute to the increase in significant ontology terms. The newer versions of MsigDB also dropped 5 out of the 10 ontology terms that were significant in the European meta-analysis. Many of the terms identified are related to biological pathways that have been implicated in PD pathogenesis such as those in autophagy, tau protein, immune cells, vesicles, and mitochondria. As recommended, we have added additional discussion on the gene ontology results:

“The gene-ontology analysis found multiple pathways that may be relevant to PD pathology (Table S6). Multiple biological processes previously implicated in PD pathogenesis such as mitochondria (response to mitochondrial depolarization), vesicles (vesicle uncoating, phagolysosome assembly, regulation of autophagosome maturation), tau protein (tau protein kinase activity), and immune cells (microglial cell proliferation, macrophage proliferation, NK T Cell differentiation)³³. Neither mitochondria or immune cell pathways were significant

in the previous European-only meta-analysis. Novel signals from the multi-ancestry approach may have given enough power to highlight these ontology terms. Out of 10 ontology terms that were significant in the previous European-only meta-analysis¹, 4 terms were not tested due to version differences in MsigDB and only 2 of the remaining terms were significant.

However the other 4 terms were still nominally significant at $P < 0.05$. This may be due to genome-wide signals that were less significant due to their heterogeneity across the different populations.”

The cell-type enrichment was performed with a mouse single cell dataset from 2018. There are now much better human datasets (e.g. Siletti et al. 2022), and it would be more informative to use this instead of the mouse dataset.

We have also run the cell-type enrichment analysis with a single cell dataset in the human midbrain (Manno et al. 2016), the most relevant brain region for PD. We found significant enrichment in dopaminergic and GABAergic neurons (Supplementary Figure S13):

Currently, close external collaborators are using the Siletti et al. 2022 dataset and have a paper with similar analyses under review at another journal. Out of respect we did not run any analysis on the newer Siletti et al. 2022 dataset.

The SMR analysis could be strengthened by running Coloc (Zuber et al. 2022). In addition,

the authors did not filter their results based on the HEIDI test, resulting in potential false positives due to LD. More importantly, the eQTL studies are all based on European samples and cannot

be used in SMR or Coloc with sumstats of a cross-ancestry GWAS. I suggest the authors to only run Coloc/SMR on the subset of their European samples.

Thank you for your suggestions. The SMR results were already run in the main European cohort. We have now clarified this in the Results section (**new additions bolded**):

“...completed Summary-based Mendelian Randomization (SMR) results **in European-only data** to correlate said genes with PD risk. “

As well as the Online Methods section:

“In total 44 genes near the novel loci were searched in a list of previously completed PD SMR results **from European-only GWAS meta analysis...**”

We have already run the analysis using the Metabrain study (see Supplementary Table S10). However, while the manuscript only discussed genes that were filtered through the HEIDI test, we neglected to remove those results in the supplementary table. We have rectified this error and apologize for this oversight.

More recent and more informative brain eQTL datasets should be used for the SMR/Coloc analysis. For example, the Metabrain study (de Klein et al. 2021), microglia eQTLs (Lopes et al. 2022, Kosoy et al. 2022), dopaminergic neurons eQTLs (Jerber et al. 2021), as well as eQTLs in multiple brain cell types (Bryois et al. 2022).

We agree these datasets are valuable resources. Our close collaborators are also using these datasets and running similar analyses on our PD GWAS results. However, the detailed analyses and interpretations are complex and beyond the scope of this paper. Out of respect for their work we are not re-analyzing these datasets.

The chromosome X was not analyzed, it would be great to have it included.

Sex is a strong predictor of PD and sex chromosomes have been underexplored in PD genomics. Unfortunately chromosome X data were not available in most ancestries and thus we did not analyze this data. We also lacked raw iDats data to recall separate sexes for data quality checks. We have tried to contact 23andMe asking for chromosome X summary statistics, but they let us know that this would require a full new collaboration and will take several months of review as well. We now have added language communicating this limitation:

“Only autosomal variants were kept in the final results as sex-chromosome data were not available for all ancestries.”

In the future we will endeavor to include the X chromosome data once they become more widely available with future releases from the Global Parkinson's Genetics Program (<https://gp2.org/>).

20.5 million variants passed QC but only 5.6 million were used in the meta-analysis. It was not clear to me why those variants were lost. Can the authors find a way to use all their QC passed SNPs or better explain why it's necessary that they lose 75% of their SNPs?

Thank you for this comment. 20.5 million variants passed QC across all cohorts and were included in the random-effect meta-analysis. However, the limitation of the MR-MEGA method means that it has a minimum cohort size threshold determined by the number of axes of genetic variation, meaning only variants present in at least 6 cohorts were kept in this specific method. We have edited the methods section to better communicate this to the reader:

“In total 20,590,839 variants met the inclusion criteria. However, MR-MEGA has a cohort-number requirement that varies based on the number of axes of variation. Therefore 5,662,641 SNPs present in at least 6 of the 7 cohorts were analyzed in the MR-MEGA analysis.”

There is a mention of PESCA in the methods section but it's unclear where it was used in the main text (if it was used).

While we had PESCA results in the supplement, we initially chose to leave out the discussion of the PESCA results from the main text as there is a large difference in cohort sizes between the European and the East Asian datasets, which can make the results difficult to interpret.

We found that lead SNPs or proxy variant ($R^2 = 1$) of *RIT2*, *BST1*, and *RIMS1* loci (rs4588066, rs6532190, and rs12528068), had high posterior probability for shared burden (0.994, 1, and 0.972) respectively. Interestingly while *RIT2* was previously found to be significant in both European-only and East-Asian-only meta-analyses, *BST1* was only nominally significant in East-Asian-only and *RIMS1* was not significant in the East-Asian study. We chose to highlight

the *RIMS1* result in the manuscript with the caveat that there is a large discrepancy between the two cohorts used in the analysis:

“Proportion of population-specific and shared causal variants (PESCA v0.3)⁹ was run for the main European and East Asian meta-analyses and all loci identified in the main analysis were explored (Supplementary Table 5). PESCA estimates the population-specific and shared burden of the variants. Lead SNP rs12528068 of *RIMS1* locus, which was significant in the European study but not in the East-Asian study, also showed high PP for shared burden at 0.972 with East-Asian specific burden at 0 and European specific burden at 0.028. We found that the identified loci generally had higher PP for shared burden than for population-specific burdens (Supplementary Table 5). However it is important to note that the sample size discrepancy between the European and East Asian data impacts our ability to ascertain population-specific burden at any of these loci.”

Figures have a poor resolution. Please increase the resolution of all figures as I am unable to review them. This unfortunately gives an impression of sloppiness which decreases my confidence in the overall analysis (in addition to the PESCA methods not being linked apparently to anything in the text).

Thank you for your suggestion and apologize for this issue. We have increased the resolutions of the figures within the manuscript and also have attached a zipped file with a high-resolution version of each of our figures.

Why are there so many genes in sup. figure 5 and sup. figure 6? It looks like multiple genes are associated per locus.

Thank you for noting this issue. Previously we annotated multiple independent signals per locus. We have modified the figures so that only one protein coding gene is highlighted per locus:

Minor:

Was LDSC run to check that samples from different cohort did not overlap? It would also be interesting to see the genetic correlation between the different datasets (within ancestry).

Thank you for your recommendation. It has been previously noted by both the developers of LDSC (<https://github.com/bulik/ldsc/wiki/FAQ#genetic-correlation-and-heritability>) and the Neale lab (<http://www.nealelab.is/blog/2017/9/20/insights-from-estimates-of-snp-heritability-for-2000-traits-and-disorders-in-uk-biobank>) that the minimum effective sample size needed for LDSC is around 5,000. Unfortunately most of our cohorts do not meet this threshold.

Nevertheless we have run LDSC and found that the genetic covariance intercepts were near 0 or close to the standard error, indicating a lack of sample overlaps. This has been added to the supplementary table 1 and we have added the following line in our Results:

“Genetic Covariance intercepts from Linkage Disequilibrium Score Regression within ancestries were close to zero or near the 95% confidence interval, implying that there is no sample overlap between the cohorts (Supplementary Table 1).”

Reporting the closest protein coding gene instead of the closest gene for the different loci would be more interesting to me as they are more likely to be functional.

Supplementary information mention ‘PD MAMA’ in multiple places. However, there is no mention of this in the main text. What does it refer to?

Table S10: why do some gene have an ensemble ID while others have a symbol? I suggest that the authors make two columns, one for the ensemble ID and one for the symbol.

Table S2: Authors should specify which is the effect allele (I guess A1).

Thank you for these suggestions. We have edited the relevant tables to report the nearest coding gene, defined in the Introduction section MAMA as “multi-ancestry meta-analysis”:

“Here we performed a large-scale multi-ancestry meta-analysis (MAMA) of PD GWASes

by including individuals from four ancestral populations: European, East Asian, Latin American, and African.”

reformatted the SMR results in table S10, and have added legends to table S2.

Again, thank you for your comments and suggestions as they have strengthened the manuscript.

Decision Letter, first revision:

16th February 2023

Dear Jeff,

Your revised Analysis "Multi-ancestry genome-wide meta-analysis in Parkinson's disease" has been seen by the original referees. You will see from their comments below that, while they find the study improved, Reviewer #2 has highlighted ongoing concerns regarding key aspects of the presentation. We remain interested in the possibility of publishing your study in Nature Genetics, but we would like to consider your response to these remaining concerns in the form of a further revision before we make a final decision on publication.

As before, to guide the scope of the revisions, the editors discuss the referee reports in detail within the team, including with the chief editor, with a view to identifying key priorities that should be addressed in revision, and sometimes overruling referee requests that are deemed beyond the scope of the current study. In this case, we ask that you carefully revise the presentation throughout for consistency and clarity, paying particular attention to the specific points noted by Reviewer #2, and that you include only putatively relevant tissues in the SMR analyses to reduce false positives. We again hope you will find this prioritized set of referee points to be useful when revising your study. Please do not hesitate to get in touch if you would like to discuss these issues further.

We therefore invite you to revise your manuscript again taking into account all reviewer and editor comments. Please highlight all changes in the manuscript text file. At this stage we will need you to upload a copy of the manuscript in MS Word .docx or similar editable format.

*2) If you have not done so already please begin to revise your manuscript so that it conforms to our Analysis format instructions, available [here](http://www.nature.com/ng/authors/article_types/index.html). Refer also to any guidelines provided in this letter.

[redacted]

We hope to receive your revised manuscript within 4-8 weeks. If you cannot send it within this time, please let us know.

Sincerely,
Kyle

Kyle Vogan, PhD
Senior Editor
Nature Genetics
<https://orcid.org/0000-0001-9565-9665>

Referee expertise:

Referee #1: Genetics, neurodegenerative diseases

Referee #2: Genetics, neuroscience, statistical methods

Reviewers' Comments:

Reviewer #1:

Remarks to the Author:

The authors took into account the points that I had raised. Even if I consider that the data available in multi-ancestry GWAS for Parkinson's disease are not yet mature enough for a truly convincing approach, I have no more specific comments on this work.

Reviewer #2:

Remarks to the Author:

I thank the authors for addressing my methodological concerns and the manuscript has improved. However, the manuscript is not easy to read due to inconsistencies between the text and figures/tables. I would really encourage the authors to carefully proofread their manuscript. I have identified a number of inconsistencies that are listed below but I am sure that many others remain.

Here are some of the issues I have identified.

The authors write: "9 of the novel loci found in the random-effect method showed homogeneous effects across the different ancestries".

However, I can count only 8 in Table S2 (No value for FASN locus). The authors should double check that they report the right number in the text (or that they did not accidentally delete an entry in the sup. table 2).

The authors write: "The IRS2 locus (lead SNP rs1078514, PANC-HET = 5.3×10^{-3}) shows the biggest departure in allelic effects in the Finnish cohort (Supplementary Figure 4)".

Do the authors mean that the effect size is 0 in the Finnish cohort? It reads as if the effect is larger in the Finnish cohort, while the plot shows the largest effect size for the AMR population.

The authors write: "The MYLK2 locus has the main non-Finnish European cohort at odds with the Latin American and Finnish cohorts(lead SNP rs6060983, PANC-HET = 0.035), suggesting different effects between populations".

I can't see this in the supplementary figure S4. It looks like the effect size is very similar between the

EUR population and the FINN population for this locus. The authors should double check that the supplementary figure matches with the text.

"Proportion of population-specific and shared causal variants (PESCA v0.3)⁹ was run for the main European and East Asian meta-analyses and all loci identified in the main analysis were explored (Supplementary Table 5). PESCA estimates the population-specific and shared burden of the variants".

I am not familiar with PESCA and don't really understand what the authors mean by 'shared burden of the variants'? Do I understand correctly that this is essentially a co-localization test to test if the local genetic architecture differs at each loci? It would be good for the authors to clarify how the method works and evaluate whether this is a useful addition to the manuscript.

In the abstract, the authors write: "By combining our results with publicly available eQTL data, we identified 23 genes near these novel loci whose expression is associated with PD risk".

The authors should tone this down (e.g. '23 putative risk genes'). SMR can lead to false positive when looking into non-relevant tissues (due to LD), or because of pleiotropy (e.g. SNP in enhancer regulating two genes but only one affecting PD risk).

Figure 2: panel labels are missing (a,b,c,d)

Table S3: Why are there more SNPs in the 99% credible set than in the 95% credible set. Did the author swap the columns?

Table S3: For rs55818311, the nearest gene/feature is indicated as SPPL2B and the nearest protein coding gene as LSM7 (both being protein-coding genes). The authors should clarify which is the closest gene (or are they equi-distant)?

It would be more readable to me to round the MAF report as percentage. For example, (e.g. AMR:0.004063). as for example AMD:0.04%.

Table S10: There are methylationQTL used in the SMR analysis that are annotated to genes. How were the methylation probes linked to a gene?.

Table S10: there is a column named 'DELETE', which I guess the authors forgot to delete.

Table S10: It would be useful to know for which locus, which gene could potentially be the risk gene according to the SMR analysis (and also know if for some locus, there are multiple candidate genes).

Table S10: Because of LD, false positives can easily arise when performing SMR into a non-disease related tissue. The authors should only use putatively relevant tissues (e.g. brain, spinal cord, gut, immune), or justify why the other tissues are relevant to PD's etiology (e.g. skin).

In the 'Data and code availability section', there is a github link for the analysis pipeline which links to a page without any code. The authors should make their code available to the community (and reviewer).

It's great that the authors are releasing the summary statistics of their study. However, I noticed that

the 'rsids' were not present in the file. I am aware that these can change between dbSNP builds and not all genetic variants have an 'rsid' but I think that a lot of people would appreciate having 'rsids' in addition to the position in the summary statistics file.

Author Rebuttal, first revision:

We would like to thank the editor for his assistance in the submission and the reviewers for their insightful comments. We have made further changes to the manuscript to address the noted inconsistencies and improve the presentation of our work.

Reviewer comments are in blue. Our responses are in black.

Reviewers' Comments:**Reviewer #1:****Remarks to the Author:**

The authors took into account the points that I had raised. Even if I consider that the data available in multi-ancestry GWAS for Parkinson's disease are not yet mature enough for a truly convincing approach, I have no more specific comments on this work.

Thank you for your comments. We hope that in the future with additional data from more diverse sources of Parkinson's disease genetics, we will be able to find results that better represent the genetic risk of Parkinson's disease at a diverse global scale.

Reviewer #2:**Remarks to the Author:**

I thank the authors for addressing my methodological concerns and the manuscript has improved. However, the manuscript is not easy to read due to inconsistencies between the text and figures/tables. I would really encourage the authors to carefully proofread their manuscript. I have identified a number of inconsistencies that are listed below but I am sure that many others remain.

Thank you for your comments and your suggestions. As suggested, we have proofread our manuscript and made additional changes that better present our findings. In addition to the changes you have recommended, we have identified additional inconsistencies including top line numbers for SMR (25 instead of 23, which was a holdover from a previous result) and gene names (*RILPL2* from *RILPL1*) and they have now been corrected. We hope that the manuscript we present to you better reflects our work.

Here are some of the issues I have identified.

The authors write: “9 of the novel loci found in the random-effect method showed homogeneous effects across the different ancestries”.

However, I can count only 8 in Table S2 (No value for *FASN* locus). The authors should double check that they report the right number in the text (or that they did not accidentally delete an entry in the sup. table 2).

Thank you for noticing the missing values for the *FASN* locus. The missing values are for the MR-MEGA results, which has a minimum cohort number requirement that the variant failed to meet. This variant was missing in both East Asian cohorts. The random-effect method found no heterogeneity across the other populations. However this may mislead our readers that this variant also has homogeneous effect in East Asian populations when this variant was actually not tested. We have made the following change to the manuscript to address this issue:

“8 of the novel loci found in the random-effect method showed homogeneous effects across the 4 different ancestries. An additional novel locus (*FASN*) identified by the random-effect method showed homogeneous effects in all available populations, but note that this variant failed QC in both East Asian datasets.”

The authors write: “The *IRS2* locus (lead SNP rs1078514, PANC-HET = 5.3×10^{-3}) shows the biggest departure in allelic effects in the Finnish cohort (Supplementary Figure 4)”.

Do the authors mean that the effect size is 0 in the Finnish cohort? It reads as if the effect is larger in the Finnish cohort, while the plot shows the largest effect size for the AMR population.

Thank you for this comment. We agree that the language may be misleading to the readers. We have updated this sentence to read:

“The *IRS2* locus (lead SNP rs1078514, $P_{\text{ANC-HET}} = 5.3 \times 10^{-3}$) shows that the effect estimate from the Finnish cohort differed most from the meta-analysis effect estimate.”

The authors write: “The *MYLK2* locus has the main non-Finnish European cohort at odds with the Latin American and Finnish cohorts(lead SNP rs6060983, $P_{\text{ANC-HET}} = 0.035$), suggesting different effects between populations”.

I can't see this in the supplementary figure S4. It looks like the effect size is very similar between the EUR population and the FINN population for this locus. The authors should double check that the supplementary figure matches with the text.

Thank you for catching this error. This has been corrected to refer to the right population:

“Similarly, the *MYLK2* locus has the African cohort at odds with the other cohorts...”

“Proportion of population-specific and shared causal variants (PESCA v0.3)⁹ was run for the main European and East Asian meta-analyses and all loci identified in the main analysis were explored (Supplementary Table 5). PESCA estimates the population-specific and shared burden of the variants”.

I am not familiar with PESCA and don't really understand what the authors mean by ‘shared burden of the variants’? Do I understand correctly that this is essentially a co-localization test to test if the local genetic architecture differs at each loci? It would be good for the authors to clarify how the method works and evaluate whether this is a useful addition to the manuscript.

Thank you for this comment. We apologize for the confusion in using the phrase “shared burden” and have re-worded it to “shared causal variants” as used in the original PESCA publication (Shi et al. AJHG 2020). You are correct, PESCA is essentially a co-localization test to compare the local genetic architecture between two populations, using GWAS

summary statistics and ancestry-specific LD estimates to account for LD differences between populations. Variants identified as shared between the populations may be more likely to be causal. In addition, we would expect the meta-analysis hits to show higher posterior probability estimates for shared causal variants than for population-specific causal variants, and even when they were not identified in the single ancestry study. This is what we observed in the *RIMS1* locus which was previously significant in the European-only study but not in the East-Asian only study. We have added the following language to better clarify the methods and the justification for using this method:

“PESCA v0.3⁹ was run for the main European and East Asian meta-analyses and all loci identified in the main analysis were explored (Supplementary Table 5).

PESCA uses ancestry-matched LD estimates to infer whether the causal variants are population-specific or shared between two populations. Variants identified as shared between the populations may be more likely to be causal. In addition, we expect higher posterior probability (PP) for shared causal variants in the loci identified by the multi-ancestry meta-analysis, even if they have not previously been identified in the single-ancestry study. The lead SNP in the *RIMS1* locus (rs12528068) had a high PP for being a shared causal variant (PP = 0.972) despite being significant in the European study¹ but not in the East-Asian study². We also observed that the novel lead variants rs35940311 (*MTF2*), rs11918587 (*PIK3CA*), rs4820434 (*EP300*), and rs60708277 (*PPP6R2*) had higher PP

estimates for being shared causal variants across both populations (PP_{shared} = 0.757, 0.214, 0.769, 0.946) than for being causal variants in a single population (PP_{EUR} < 0.080, PP_{EAS} < 0.001). However it is important to note that the sample size discrepancy between the European and East Asian data impacts our power to detect population-specific causal variants at any of these loci.”

In the abstract, the authors write: “By combining our results with publicly available eQTL data, we identified 23 genes near these novel loci whose expression is associated with PD risk”. The authors should tone this down (e.g. ‘23 putative risk genes’). SMR can lead to false positive when looking into non-relevant tissues (due to LD), or because of pleiotropy (e.g. SNP in enhancer regulating two genes but only one affecting PD risk).

Thank you for your suggestion. We agree with the language change and we have made the change in the abstract below:

“By combining our results with publicly available eQTL data, we identified 25 putative risk genes near these novel loci whose expression is associated with PD risk.”

Figure 2: panel labels are missing (a,b,c,d)

Thank you for noticing this error. We have now added the correct labels:

Table S3: Why are there more SNPs in the 99% credible set than in the 95% credible set. Did the author swap the columns?

Thank you for this comment. Mathematically, 99% credible sets must have at least as many SNPs as 95% credible sets because more SNPs are necessary to achieve a cumulative sum of 99% posterior probability.

Table S3: For rs55818311, the nearest gene/feature is indicated as SPPL2B and the

nearest protein coding gene as LSM7 (both being protein-coding genes). The authors should clarify which is the closest gene (or are they equi-distant)?

Thank you for noticing this discrepancy. We have properly assigned *SPPL2B* as the nearest protein coding gene.

It would be more readable to me to round the MAF report as percentage. For example, (e.g. AMR:0.004063). as for example AMD:0.04%.

Thank you for your suggestion. We have made this change in the Results section as follows:

“The lead SNPs (rs578139575 and rs73919910) for these loci are non-coding and very rare in European populations but more common in Africans and Latin Americans (gnomAD v3.1.2 allele frequencies in EUR: 0.02%, 0.23%; AFR: 1.64%, 8.84%; AMR: 0.41%, 1.91%).”

And throughout the manuscript including the Table 2 of novel loci:

rsID	Nearest Coding Gene	SMR nominated putative genes	CHR:BP:A1:A2	BETA(REF)	SE	P (REF)	P(MR-MEGA)	P(ANCE-HET)	P(RES-HET)	gnomAD EUR AF	gnomAD EAS AF	gnomAD AMR AF	gnomAD AFR AF	pLI
rs11164870	MTF2	CCDC18	1:93552187:C:G	0.054	0.009	1.15E-10	2.64E-09	0.229	0.928	39.02%	35.05%	45.16%	85.04%	1
rs6806917	PIK3CA	KCNMB3	3:178861417:T:C	-0.070	0.011	1.65E-10	3.43E-09	0.215	0.762	82.03%	89.92%	77.46%	57.83%	1
rs16843452	ADD1	AD D1, NOP14- AS1, NOP14	4:2849168:T:C	-0.068	0.012	4.11E-09	3.19E-07	0.747	0.687	18.45%	47.37%	18.19%	8.92%	0.99
rs6469271	SYBU	SYBU	8:110644774:T:C	-0.056	0.010	3.62E-09	2.04E-07	0.590	0.954	77.49%	59.34%	74.69%	61.46%	0
rs107851	IRS2	None	13:110463168:	0.068	0.02	0.00481	2.30E-5.30E-	0.261		33.28%	39.20%	40.56	10.72%	0.9

4			T:C	6	7	09	03							9
rs28648524	USP8	AC084756, TRPM7	15:50787409:A:T	0.064	0.010	6.45E-10	2.58E-08	0.406	0.661	78.13%	53.73%	76.50%	79.78%	1
rs11650438	PIGL	ADORA2B, ZSWIM7, PIGL, TTC19, NCOR1, CENPV, TRPV2	17:16234260:A:G	0.050	0.009	2.93E-09	1.46E-07	0.528	0.288	46.94%	17.83%	48.49%	64.02%	0
rs4485435	FASN	None	17:80045086:C:G	0.082	0.014	2.61E-09	N/A	N/A	N/A	17.26%	12.05%	34.76%	30.27%	1
rs6060983	MYLK2	None	20:30420924:T:C	0.069	0.037	0.03221	3.86E-09	0.035	0.149	69.26%	98.95%	71.81%	29.00%	0.23
rs1736020	USP25	None	21:16812552:A:C	0.006	0.005	0.8853	1.12E-09	4.74E-05	0.638	42.97%	18.58%	38.64%	13.18%	0.75
rs73174657	EP300	ZC3H7B, POLR3H, CSDC2, PMM1, RANGAP1, MEI1, L3MBTL2, PMM1, SLC25A17	22:41434158:A:G	-0.059	0.010	3.81E-09	4.90E-07	0.983	0.655	27.23%	6.30%	47.51%	14.17%	1
rs10775809	PPP6R2	PPP6R2	22:50808017:A:T	0.092	0.015	4.09E-10	5.61E-08	0.943	0.903	10.10%	80.31%	80.13%	56.54%	0.16

Table S10: There are methylationQTL used in the SMR analysis that are annotated to genes. How were the methylation probes linked to a gene?.

Thank you for raising this question. We have annotated probes with annotation data from IlluminaHumanMethylation450kanno.ilmn12.hg19, an annotation data package in Bioconductor specialized for Illumina 450k methylation array. We have re-run the annotation and there were minimal changes to the results (cg00949728: *NOP14-AS1,NOP14* to *NOP14*; cg03131358: *MEI1* to *CCDC134*; cg17742451 *PIGL,CENPV* to *CENPV*; cg06805925 to *PIGL* to none). We have further specified this in the legend section of the Supplementary Table S10:

“HGNC gene name of the gene of interest. In case of methylation data, the methylation probe was annotated using data from Bioconductor R package

“IlluminaHumanMethylation450kanno.ilmn12.hg19.”

Table S10: there is a column named ‘DELETE’, which I guess the authors forgot to delete.

Thank you for noticing this error. The column has been deleted.

Table S10: It would be useful to know for which locus, which gene could potentially be the risk gene according to the SMR analysis (and also know if for some locus, there are multiple candidate genes).

Thank you for this suggestion. We have edited Table 2 to include the SMR nominated putative genes (please see above).

Table S10: Because of LD, false positives can easily arise when performing SMR into a non-disease related tissue. The authors should only use putatively relevant tissues (e.g. brain, spinal cord, gut, immune), or justify why the other tissues are relevant to PD’s etiology (e.g. skin).

Thank you for your suggestion. We have further filtered our SMR results to only include tissues in the central nervous system, digestive system, and blood. We have now stated this in the online methods section manuscript:

In total, 44 genes near the novel loci were searched in a list of previously completed PD SMR results from European-only GWAS meta-analysis^{17,48–56}.

Only

tissues in the central nervous system, digestive system, and blood were used due to their relevance to PD pathology.

In the ‘Data and code availability section’, there is a github link for the analysis pipeline which links to a page without any code. The authors should make their code available to the community (and reviewer).

Thank you for your interest in open science. We have now made the code for the meta-analysis and several downstream analyses public in the repository.

It's great that the authors are releasing the summary statistics of their study. However, I noticed that the 'rsids' were not present in the file. I am aware that these can change between dbSNP builds and not all genetic variants have an 'rsid' but I think that a lot of people would appreciate having 'rsids' in addition to the position in the summary statistics file.

Thank you for raising this point. We agree that rsID would be a useful addition to the summary statistic. We have added rsID to the summary statistic and have updated the Google Drive link to the summary statistic with the rsID:

<https://drive.google.com/file/d/1nCikCdD5NI9L8EDmpMJHvUSvWkno5Ulw/>

Decision Letter, second revision:

8th May 2023

Dear Jeff,

Your revised manuscript "Multi-ancestry genome-wide meta-analysis in Parkinson's disease" (NG-A60871R1) has been seen by Reviewer #2. As you will see from the comments below, Reviewer #2 is satisfied with the revision and has no remaining requests, and therefore we will be happy in principle to publish your study in Nature Genetics as an Article pending final revisions to comply with our editorial and formatting guidelines.

We are now performing detailed checks on your paper, and we will send you a checklist detailing our editorial and formatting requirements soon. Please do not upload the final materials or make any revisions until you receive this additional information from us.

Thank you again for your interest in Nature Genetics. Please do not hesitate to contact me if you have any questions.

Sincerely,
Kyle

Kyle Vogan, PhD
Senior Editor
Nature Genetics
<https://orcid.org/0000-0001-9565-9665>

Reviewer #2 (Remarks to the Author):

The authors have now addressed all my concerns. I feel that the manuscript is now much easier to read, and I find the results of the study exciting.

Reviewer #2 (Remarks to the Author):

Author Rebuttal, second revision:

The authors have now addressed all my concerns. I feel that the manuscript is now much easier to read, and I find the results of the study exciting.

Thank you for your comments as they have improved the legibility and the content of our work. We share your excitement and hope to share the final manuscript with the scientific community soon.

Final Decision Letter:

20th Oct 2023

Dear Jonggeol,

I am delighted to say that your manuscript "Multi-ancestry genome-wide association meta-analysis of Parkinson's disease" has been accepted for publication in an upcoming issue of Nature Genetics.

Your paper will be published online after we receive your corrections and will appear in print in the next available issue. You can find out your date of online publication by contacting the Nature Press Office (press@nature.com) after sending your e-proof corrections. Now is the time to inform your Public Relations or Press Office about your paper, as they might be interested in promoting its publication. This will allow them time to prepare an accurate and satisfactory press release. Include your manuscript tracking number (NG-A60871R2) and the name of the journal, which they will need when they contact our Press Office.

Please note that *Nature Genetics* is a Transformative Journal (TJ). Authors may publish their research with us through the traditional subscription access route or make their paper immediately open access through payment of an article-processing charge (APC). Authors will not be required to make a final decision about access to their article until it has been accepted. [Find out more about Transformative Journals](https://www.springernature.com/gp/open-research/transformative-journals)

Authors may need to take specific actions to achieve [compliance with funder and institutional open access mandates](https://www.springernature.com/gp/open-research/funding/policy-compliance-faqs). If your research is supported by a funder that requires immediate open access (e.g. according to [Plan S principles](https://www.springernature.com/gp/open-research/plan-s-compliance)) then you should select the gold OA route, and we will direct you to the compliant route where possible. For authors selecting the subscription publication route, the journal's standard licensing terms will need to be accepted, including [self-archiving-and-license-to-publish](https://www.nature.com/nature-portfolio/editorial-policies/self-archiving-and-license-to-publish). Those licensing terms will supersede any other terms that the author or any third party may assert apply to any version of the manuscript.

To assist our authors in disseminating their research to the broader community, our SharedIt initiative provides you with a unique shareable link that will allow anyone (with or without a subscription) to read the published article. Recipients of the link with a subscription will also be able to download and

print the PDF.

If you have not already done so, we invite you to upload the step-by-step protocols used in this manuscript to the Protocols Exchange, part of our on-line web resource, [natureprotocols.com](https://www.nature.com/natureprotocols). If you complete the upload by the time you receive your manuscript proofs, we can insert links in your article that lead directly to the protocol details. Your protocol will be made freely available upon publication of your paper. By participating in [natureprotocols.com](https://www.nature.com/natureprotocols), you are enabling researchers to more readily reproduce or adapt the methodology you use. [Natureprotocols.com](https://www.nature.com/natureprotocols) is fully searchable, providing your protocols and paper with increased utility and visibility. Please submit your protocol to <https://protocolexchange.researchsquare.com/>. After entering your [nature.com](https://www.nature.com) username and password you will need to enter your manuscript number (NG-A60871R2). Further information can be found at <https://www.nature.com/nature-portfolio/editorial-policies/reporting-standards#protocols>

Sincerely,
Wei

Wei Li, PhD
Senior Editor
Nature Genetics
New York, NY 10004, USA
www.nature.com/ng